# Therapeutic Targets in Diffuse Midline Gliomas—An Emerging Landscape

**DOI:** 10.3390/cancers13246251

**Published:** 2021-12-13

**Authors:** Elisha Hayden, Holly Holliday, Rebecca Lehmann, Aaminah Khan, Maria Tsoli, Benjamin S. Rayner, David S. Ziegler

**Affiliations:** 1Children’s Cancer Institute, Lowy Cancer Research Centre, UNSW Sydney, Kensington 2052, Australia; ehayden@ccia.org.au (E.H.); HHolliday@ccia.org.au (H.H.); RLehmann@ccia.org.au (R.L.); AKhan@ccia.org.au (A.K.); mtsoli@ccia.org.au (M.T.); brayner@ccia.org.au (B.S.R.); 2School of Women’s and Children’s Health, Faculty of Medicine, University of New South Wales, Kensington 2052, Australia; 3Kids Cancer Centre, Sydney Children’s Hospital, Randwick 2031, Australia

**Keywords:** diffuse midline gliomas, molecular targets, potential therapy development

## Abstract

**Simple Summary:**

Diffuse midline gliomas (DMGs) remain one of the most devastating childhood brain tumour types, for which there is currently no known cure. In this review we provide a summary of the existing knowledge of the molecular mechanisms underlying the pathogenesis of this disease, highlighting current analyses and novel treatment propositions. Together, the accumulation of these data will aid in the understanding and development of more effective therapeutic options for the treatment of DMGs.

**Abstract:**

Diffuse midline gliomas (DMGs) are invariably fatal pediatric brain tumours that are inherently resistant to conventional therapy. In recent years our understanding of the underlying molecular mechanisms of DMG tumorigenicity has resulted in the identification of novel targets and the development of a range of potential therapies, with multiple agents now being progressed to clinical translation to test their therapeutic efficacy. Here, we provide an overview of the current therapies aimed at epigenetic and mutational drivers, cellular pathway aberrations and tumor microenvironment mechanisms in DMGs in order to aid therapy development and facilitate a holistic approach to patient treatment.

## 1. Introduction

Diffuse midline gliomas (DMG), including diffuse intrinsic pontine gliomas (DIPGs), are aggressive central nervous system (CNS) pediatric tumours located in the brainstem, thalamus, spinal cord and cerebellum [1,2]. Their prognosis is dismal and patients have a five-year survival of less than 1% due to their high level of resistance to current standard therapies, with no significant improvement in the treatment or prognosis of DMGs having occurred in more than three decades [3]. DMG tumours typically display no contrast enhancement by magnetic resonance imaging (MRI), suggesting an intact blood–brain barrier (BBB) that impedes further the delivery of therapeutic agents [4]. As such, new and innovative treatment strategies are urgently needed to counter these devastating tumours. To facilitate research, a greater understanding of the underlying molecular mechanisms that promote DMG tumour growth is required. Approximately 80% of DMGs harbour lysine-to-methionine substitutions at position 27 in Histone 3 genes encoding H3.3 (*H3F3A*), and to a lesser extent H3.1 (*HIST1H3B*)*,* collectively referred to as H3K27M [5,6,7]. In 2016 the World Health Organization (WHO) classified all gliomas harbouring the H3K27M mutant as a new entity, a “Diffuse Midline Glioma H3 K27M-mutant” [1]. Subsequent functional studies demonstrated that the H3K27M mutation drives DMG growth, suggesting that this epigenetic dysregulation is the key promotor of DMG tumorigenesis through the global reduction in the repressive epigenetic mark H3K27me3 [8,9,10]. In 2021 the WHO further classified the inclusion of a subset of DMGs that lack the H3K27M mutation but exhibit a global loss of tri-methylation [2], potentially mediated by the overexpression of EZH inhibitory protein (EZHIP) which functionally acts as a K27M-like inhibitor of polycomb repressive complex 2 (PRC2) [11]. Collectively, these tumors have now been termed “diffuse midline glioma, H3 K27-altered”. Combined, this knowledge has resulted in research focussed on the development of pharmacological inhibitors designed to regulate these epigenetic mechanisms by particularly targeting epigenetic modifiers, including methyltransferase activity [12,13], demethylases [14,15], acetylation [16], chromatin readers and writers [17,18,19], and histone chaperones [20,21]. It is also becoming increasingly apparent that there exists a link between DMG cell metabolism and the underlying activation status of key epigenetic marks in DMGs, resulting in novel DMG targets involving metabolic pathway regulation [22,23,24]. Furthermore, the developed understanding of DMG tumour biology has also revealed additional novel therapeutic targets within the tumour microenvironment, particularly within the regulation of the immune system [25,26] and the interplay of DMG tumours with neuronal cells [27]. This comprehensive review summarises the current understanding of DMG biology in relation to therapeutic targets in preclinical development and proposes further research avenues to highlight novel mechanisms capable of being manipulated for the treatment of DMGs.

## 2. Targeting Epigenetic Mechanisms in DMGs

Since the discovery of the importance of the H3K27M mutation to DMG tumorgenicity a plethora of inhibition studies within this disease setting have been undertaken, summarised in Table 1. The H3K27M mutation causes broad epigenetic reprogramming through the inhibition of PRC2, which deposits the repressive histone mark H3K27me3. This results in global H3K27 hypomethylation in all histone H3 variants and is central to DMG oncogenesis [5,6,7,28,29]. Restoring H3K27me3, therefore, is a key therapeutic strategy to reduce DMG tumorigenicity, and targeting the enzymes that erase this mark—histone demethylases—is one such way to achieve this. The demethylation of H3K27me3 is catalysed by the lysine demethylase 6 (KDM6) subfamily of demethylases, consisting of JMJD3 and UTX [30]. Confirmation of their role in DMGs was obtained through the use of GSKJ4, a pro-drug for GSKJ1, which is a potent and selective pharmacological inhibitor of JMJD3 and UTX [31]. Hashizume et al. found that GSKJ4 treatment in vitro increased H3K27me3 levels and reduced cell viability as well as clonogenicity in H3K27M-mutant DIPG cells but not H3-WT DMGs or normal astrocytes [32]. The genetic depletion of JMJD3, but not UTX, was able to phenocopy GSKJ4 treatment, indicating that JMJD3 is the enzyme target responsible for demethylating H3K27me3 in DMGs [32].

Similarly, GSKJ4 treatment significantly extended survival in two H3K27M-mutant DMG PDX models but had no effect in an H3-WT glioma PDX model. The active derivative of the drug, GSKJ1, was detected in the brainstem of non-tumour-bearing mice, confirming BBB penetration of the drug [32]. Delving further into the transcriptional consequences of GSKJ4, Hashizume et al. found that treatment decreased the chromatin accessibility and expression of genes involved in DNA double-strand break repair, such as *PCNA* and *XRCC1*. Moreover, GSKJ4 was able to enhance radiation-induced DNA damage and subsequent apoptosis in H3K27M-mutant DIPGs both in vitro and in vivo [33]. Building upon these findings, combining GSKJ4 with APR-246, an inhibitor of mutant TP53, further enhanced radiosensitivity in vitro [34]. GSKJ4 is not yet in clinical development due to challenges associated with the rapid conversion of the pro-drug into its active form, which has restricted cell permeability due to its high polarity, indicating that efforts should be therefore focused on increasing the stability of GSKJ4 in vivo if it were to progress from a potential to a beneficial treatment for DMGs. 

Importantly, the inhibition of PRC2 by H3K27M is not equivalent to complete PRC2 loss of function, and despite a global reduction in H3K27me3 in DIPGs PRC2 and its product, H3K27me3, persist at hundreds of genomic loci. This occurs at CpG islands, high-affinity PRC2 binding sites thought to be “strong” PRC2 targets [12,13,17]. Genes that remain epigenetically silenced in DIPGs by PRC2 include *CDKN2A,* encoding the cell cycle inhibitor p16, and neuronal differentiation genes [13,17]. Enhancer of zeste homolog 2 (EZH2) is the histone methyltransferase catalytic subunit of PRC2. Targeting residual PRC2 activity with EZH2 inhibitors to switch on tumour suppressor and pro-differentiation genes is another plausible avenue for DMG treatment. Indeed, treatment of multiple DMG cell lines with the EZH2 inhibitors GSK343 and EPZ6438 reduced H3K27me3 at the *CDKN2A* locus and reactivated the expression of p16, accompanied by a reduction in the proliferation and induction of cellular senescence [13]. Importantly, H3-WT glioblastoma (GBM) lines, or those with G34R mutations, were not affected by EZH2 inhibitors, demonstrating the specific H3K27M context required for EZH2 inhibitor efficacy [13]. In contrast, Wiese et al. found no cytotoxic effect of EPZ6438 in DIPG models regardless of H3K27M status [35]. This discrepancy could be due to different time points selected for their proliferation assays (15 days versus 3 days). The genetic knockout of EZH2 prolonged survival when H3K27M mouse tumour cells were orthotopically transplanted, demonstrating that EZH2 activity is required for tumour growth in vivo [13]. In agreement, the knockdown of other core components of the PRC2 complex, EED and SUZ12, reduced the growth and survival of human DIPG cells in vitro [17]. This was mediated via p16 de-repression in the SF8628 model, but not for SU-DIPG-VI cells, indicating that p16-independent mechanisms are also important for PRC2-mediated tumorigenesis. 

The above studies demonstrate therapeutic potential for targeting residual PRC2 activity in H3K27M-mutant DIPGs. EPZ6438 (Tazemetostat) is currently being trialled in other solid tumours such as pediatric sarcomas (NCT02601937) and lymphoma (NCT01897571). However, there are limited data on the activity of EZH2 inhibitors in animal models of DIPGs due to their poor BBB penetration. While brain delivery of EZH2 inhibitors can be improved by co-administration of the drug efflux transporter inhibitor elacridar, its clinical development has been discontinued [46]. Further pre-clinical testing in orthoptic mouse models is required before EZH2 inhibitors can be considered for clinical applications in DMG.

Polycomb repressive complex 1 (PRC1) is another repressive polycomb complex which plays critical roles during embryonic development and catalyses the ubiquitination of histone H2A at lysine 119 (H2AK119ub) [47]. BMI1 is a core component of PRC1 and acts as an oncogene in several cancers, including DMGs, where it promotes self-renewal and stem cell maintenance [36,37]. Expression of BMI1 and its associated mark, H2AK119ub, is upregulated in DIPGs compared to normal pons [37]. BMI1 expression is directly regulated by H3K27M, as ectopic expression of the oncohistone in H3-WT cells caused increased BMI1 and H2AK119ub levels and increased proliferation, while the inverse was observed when WT H3.3 was expressed in H3K27M-mutant DMG cells. This was reflected on the chromatin level, where the removal of H3K27M increased H3K27me3 at the *BMI1* locus [37]. Targeting BMI1 with small-molecule inhibitors PTC209 and PTC028 was effective against H3K27M-mutant DIPG models in vitro and in vivo, reducing global BMI1 chromatin binding. However, long-term drug exposure resulted in recurrence due to an accumulation of secretory senescent cells. Combining the BMI1 inhibitor PTC028 (oral gavage 12.5 mg/kg) with the BH3 protein mimetic obatoclax (IP 3 mg/kg) significantly increased survival time compared to BMI1 inhibition alone in an orthotopic DMG PDX model, mediated through the clearing of residual senescent DMG cells by obatoclax. The inhibition of BMI1 with another drug, PCT596, was also shown to be effective both in vitro and in vivo [38]. A clinical trial for PTC596 in DMG and HGG patients is currently recruiting (NCT03605550). Given the recent data revealing the potentially pro-tumorigenic activity of prolonged BMI1 inhibition, caution will be needed in clinical translation and the selection of dosing schedules. 

H3K27M-mutant DMGs have elevated levels of H3K27ac, an active epigenetic mark found at the promoters and enhancers of actively transcribed genes [28]. The enzymes responsible for writing and erasing histone acetylation are histone acetyltransferases (HATs) and histone deacetylases (HDACs), respectively. Histone acetylation is read by the bromodomain and extra terminal (BET) family of proteins, including BRD4. HDAC inhibitors are perhaps the most studied mode of epigenetic-targeted therapy in DMGs. These drugs increase histone acetylation by targeting enzymes that remove the mark. Grasso et al. identified HDAC inhibitors from a targeted drug screen performed across a panel of human DIPG cultures. Panobinostat, a pan-HDAC inhibitor, was selected for follow-up due to its higher potency compared to other HDAC inhibitors [16]. Panobinostat treatment caused a dose-dependent increase in histone acetylation and, unexpectedly, a partial rescue of H3K27me3. Panobinostat displayed in vitro efficacy, reducing DMG cell proliferation and inducing cell death, and was also efficacious in vivo in both H3K27M-mutant and H3-WT DMG models when delivered systemically (IP 20 mg/kg once a week for 4 weeks), reaching doses in the brain higher than in vitro IC50. However, when surviving mice were re-challenged with panobinostat, DMG tumours were found to be resistant, highlighting a need for combination therapies [16]. In another study, panobinostat at the same dose was not well-tolerated over an extended treatment, resulting in significant toxicity and no survival benefit [39]. Similarly a phase I trial of panobinostat in DMGs has shown significant toxicity, limiting dose escalation with no efficacy yet demonstrated [9]. Together these data suggest that although panobinostat may have some anti-DMG activity it is limited by a narrow therapeutic window, suggesting that combination therapies may be needed to lead to clinical efficacy.

Given that H3K27ac is already elevated in DMGs, it seems counter-intuitive that further elevation of it through HDAC inhibition can reduce DMG growth. There are two suggested explanations for this, which are not necessarily mutually exclusive. In the first, panobinostat-mediated poly-acetylation of residues nearby the mutated K27M residue disrupt the repression of PRC2, thereby restoring H3K27me3 and diminishing DMG growth [16,48]. In the second, panobinostat causes histone hyperacetylation and aberrant transcription of endogenous retroviruses, triggering an interferon response which sensitises cells to the innate immune system [42]. Panobinostat has been tested in combination with a suite of other epigenetic-targeted therapies in DIPGs. The histone demethylase inhibitor GSKJ4 demonstrated in vitro synergy with panobinostat [16], warranting further in vivo testing and mechanistic studies. The disruption of oncogenic transcription using either THZ1 or JQ1 was also synergistic with panobinostat in vitro [40]. CDK7, the target of THZ1, phosphorylates RNA polymerase II (Pol II); this is required for transcription. Mechanistically, THZ1 and panobinostat reduced H3K27ac at super-enhancer-associated genes fundamental to DMG biology, such as those important for communication with neurons in the microenvironment [40,49]. Importantly, panobinostat-resistant cells also became resistant to BET inhibitor JQ1, likely due to their shared transcriptional targets [40]. Krug et al. found that the combination of panobinostat with the DNA demethylating agent 5-azacytidine prolonged survival in two DMG PDX models [42]. Finally, another potent panobinostat combination is with CBL0137, which targets the histone chaperone facilitates chromatin transcription (FACT) [21], discussed in more detail below. 

Bifunctional inhibitors are an innovative approach for the simultaneous targeting of epigenetic modifiers. Corin is one such example of these hybrid drugs, derived from the HDAC inhibitor entinostat and an LSD1 inhibitor [45]. LSD1 demethylates the active enhancer mark H3K4me1. LSD1 inhibitors have also recently been suggested to enhance immune sensitivity in DMGs [50]. Corin treatment increased H3K27ac, H3K4me1 and H3K27me3, reduced proliferation and induced apoptosis as well as neuronal differentiation in vitro, and reduced DMG PDX growth when delivered by convection-enhanced delivery (CED) [45]. Thus, HDAC inhibitors hold promise as future DMG therapeutics, especially in combination with other epigenetic drugs. 

The bromodomain and extra-terminal (BET) family of proteins, consisting of BRD2, BRD3, BRD4 and BRDT, are chromatin readers and positive regulators of transcription. BET proteins bind to acetylated histones via their bromodomains and recruit the transcription elongation complex to promote Pol-II-mediated transcription of oncogenes such as *MYC* [51,52]. BRD4 is the predominant, and consequently most studied, member of the BET family of proteins [52]. In DMGs, H3K27M forms heterotypic nucleosomes with H3K27ac which are bound by BRD4 at super-enhancers, large clusters of enhancers with high levels of transcription factor binding, controlling the expression of cell identity genes [17]. Genes associated with BRD4-bound super-enhancers in cancer are enriched for oncogenic drivers [18]. Consequently, DMG cells are particularly vulnerable to transcriptional disruption using BET inhibitors [40]. 

JQ1, a well-studied BET inhibitor that competitively binds to bromodomains [53], is a potent inhibitor of DMG growth in vitro [17,19,40]. H3K27M-mutant cell lines were more sensitive to JQ1 than H3-WT glioblastoma cells [17]. JQ1 caused growth arrest, in part by reducing *MYC* transcription, but its effects on apoptosis were minimal [17,40]. Rather, this drug increased the expression of mature neuronal marker genes (*TUJ1* and *MAP2*) and induced neuronal-like morphological changes, suggesting that JQ1 is a differentiation therapy in DIPG [17]. Mechanistically, BET inhibition resulted in an overall shutdown of transcription [17] and reduced promoter–enhancer looping at tumour-specific genes (e.g., *OLIG2* and *SOX6*) [41]. The genetic depletion of BRD4 resulted in a stark decrease in tumour growth and improved survival when xenografted into mice [40], demonstrating the dependence of DIPG tumours on BRD4. Consistently, treatment of orthoptic DMG PDXs with JQ1 (IP 50 mg/kg daily for 10 days) reduced tumour burden and significantly extended survival [17]. However, when other BET inhibitors were tested, namely iBET762 and OTX015, they were less potent than JQ1 in vitro and were not able to achieve sufficient levels in the brain [40]. 

One of the key advantages of synergistic combination therapies is that the dose of each monotherapy can be reduced, potentially circumventing the drug delivery issues seen with BET inhibitors. Several BET inhibitor combinations have been explored in DMGs. Nagaraja et al. combined JQ1 with both panobinostat (HDAC inhibitor) and THZ1 (CDK7 inhibitor) and observed synergy with both epigenetic drugs in three human DMG cultures in vitro. However, as mentioned above, DMG cells had shared resistance to both panobinostat and JQ1 due to their similar transcriptional targets [40]. The combination of JQ1 and EZH2 inhibitor EPZ6438 blocked proliferation, increased apoptosis and extended survival in a mouse model of DMGs [43]. Given the poor BBB penetration of EPZ6438 [46], it is surprising that this drug was able to reduce tumour burden. Further preclinical testing of dual BET and EZH2 targeting is required to see if this promising result is consistent across additional DMG models, including human-derived models. Finally, JQ1 has been combined with ICG-001, which targets CREB-binding protein (CBP)—a histone acetyltransferase (HAT) that also interacts with other transcriptional regulators, such as BRD4 itself [44]. JQ1 and ICG-001 reduced in vitro proliferation, self-renewal and migration, and increased radiosensitivity [44]. While ICG-001 has not yet been tested in orthotopic models, this drug is interesting proposed to increase BBB permeability by targeting endothelial cells [54]. Therefore, co-treatment with ICG-001 may both improve the delivery of, as well as synergise with, JQ1.

While the effect of JQ1 on DMGs has been immensely beneficial for understanding the biology of BET proteins and establishing BET inhibition as an anti-cancer strategy, this compound has a poor pharmacokinetic profile. Other BET inhibitors, including OTX015, CPI-0610 and iBET762, have been tested in clinical trials for a variety of cancers (see [55] for a comprehensive review). It is important to acknowledge that significant toxicities such as gastrointestinal disorders, anaemia and thrombocytopenia were reported in some of these trials, stressing the need to find potent synergies and to define predictive biomarkers of response. 

The FACT histone chaperone complex is important for maintaining nucleosome stability and the local recycling of histones during transcription, replication and repair [56,57]. It is required in large amounts in transcriptionally active cells, with higher expression in cancer and stem cells compared to normal, differentiated cells [58,59]. Accordingly, cancer cells are exquisitely sensitive to curaxins, which are indirect inhibitors of the FACT complex. Curaxins are derived from anti-malarial quinacrine drugs and have been screened for their ability to simultaneously activate p53 and suppress NF-κB-mediated transcription without causing genotoxicity [20]. Their proposed mechanism is through DNA intercalation and the subsequent unfolding of nucleosomes, causing FACT to become trapped in chromatin and thereby depleting the pool of functionally active soluble FACT [20,59,60]. CBL0137 is the lead curaxin in clinical development due to its high stability and water solubility [20].

It has recently been found that DMG growth is inhibited by targeting FACT with CBL0137 or by the shRNA-mediated knockdown of FACT subunits SPT16 and SSRP1. CBL0137 had higher efficacy in H3K27M mutants compared to H3-WT models and minimal toxicity to normal cells, which express much lower levels of FACT [21]. CBL0137 administered as a single agent (IV 50 mg/kg once a week for 4 weeks or 20 mg/kg 5 days on/2 days off for 3 weeks) significantly extended the survival of mice bearing orthotopic DIPG tumours; the drug reached clinically relevant doses in the brain [21]. Importantly, FACT directly interacts with H3K27M and FACT inhibition increased H3K27me3, indicating that FACT is required for H3K27M’s oncogenic effects [21]. In support of this, the depletion of FACT can scramble histone positioning due to the loss of local nucleosome recycling [61,62]. It seems that FACT is required for maintaining H3K27M on chromatin; inhibition with CBL0137 causes H3K27M to be ejected or mis-localised, removing its downstream epigenetic effects. Furthermore, CBL0137 potently synergised with panobinostat both in vitro and in vivo and was accompanied by an increase in both H3K27me3 and H3K27ac [21]. Further investigation will be important for dissecting the precise genomic loci affected by these epigenetic therapies. In addition, combination treatment suppressed the expression of cell cycle genes (e.g., *E2F1* and *CDK4*) and oligodendrocyte developmental genes (e.g., *ASCL1* and *LINGO1)* [21]. Such lineage genes are known to be regulated by super-enhancers in DMGs and are disrupted by treatment with panobinostat [40]. It is therefore possible that CBL0137 exacerbates the effects of panobinostat on the super-enhancer landscape in DIPGs. CBL0137 has recently completed phase I clinical trials in adults with recurrent solid tumours and was found to be well-tolerated as well as exhibit preliminary anti-tumour activity (NCT01905228) [63]. Excitingly, this drug is due to enter a phase I/II trial (NCT04870944) in paediatric cancer patients, with activity to be tested in a DMG expansion cohort.

In addition to histone hypomethylation in H3K27M-mutant DMGs, these tumours are also characterised by global DNA hypomethylation, which is thought to contribute to the highly characteristic gene expression program in DIPGs [29,64]. Genome-wide methylation profiling in GBMs revealed that H3K27M GBMs had a distinct methylome to H3-WT GBMs, with differential methylation of genes involved in neuronal development [64,65]. Given the intimate link between H3K27me3 and DNA methylation [66], it follows that H3K27M, which hinders H3K27me3, would also impact DNA methylation. DNA methylation primarily occurs on the fifth position of cytosine (5-methylcytosine; 5mC) in the CpG sequence context and is generally associated with gene silencing [67,68]. DNA methylation is mediated by DNA methyltransferase (DNMT) enzymes [68]. DNA demethylating agents, such as 5-azacytidine (5-aza) and decitabine, are nucleoside analogues that target DNMT activity. Treatment with 5-aza improved survival in two DMG PDX models as a single agent (IP 3 mg/kg 5 days on/2 days off for 4 weeks), and its efficacy was further enhanced when combined with panobinostat (IP 10 mg/kg) [42]. The mechanism involved further loss of DNA methylation and, similar to HDAC inhibition, subsequent de-repression of silent endogenous retroviruses, triggering an interferon response and viral mimicry [42]. Testing 5-aza and panobinostat in immunocompetent mouse models is warranted to investigate if this translates to an augmented immune response. DNA-demethylating agents are commonly used in chronic myelomonocytic leukaemia (CMML), acute myeloid leukaemia (AML) and myelodysplastic syndrome, and are currently in clinical trials in solid tumours, including adult HGGs (NCT03666559). 

The active removal of DNA methylation involves ten-eleven translocation (TET) enzymes, which catalyse the conversion of 5 mC into 5-hydroxymethylcososine (5hmC) [69]. Thymine DNA glycosylases and base excision repair machinery further process and excise 5 hmC to restore unmethylated cytosines [70]. DMG tumours have elevated mRNA expression of *TET1* and *TET3* compared to a patient-matched normal brain in addition to increased levels of 5 hmC compared to paediatric GBMs [71]. These findings hint that active DNA demethylation by TET enzymes contributes to DMGs’ hypomethylated signature and pathological gene expression. In addition, similar to histone demethylase enzymes, TET enzymes are dependent on the metabolic intermediate α-KG as a co-factor. It is therefore possible that the increased α-KG levels in DMGs [22] fuel TET-mediated DNA demethylation to maintain a hypomethylated genome. Thus, targeting TET enzymes to restore DNA methylation may represent a novel avenue of treatment for DMGs. The TET inhibitor Bobcat339 has recently been synthesised as a tool compound for understanding TET biology and is a promising starting point for the development of therapeutic inhibitors of DNA demethylation [72].

Similarly, investigation into the RNA polymerase II (RNAPII) transcriptional machinery has gained traction in recent years [18,73]. This includes targeting CDK7 in order to prevent RNAPII phosphorylation and subsequent transcription initiation, with the CDK7 inhibitor THZ1 having been shown to suppress tumour growth in orthotopic DMG orthotopic [74]. More recently, CDK9 suppression has demonstrated anti-tumour effects in DMG models [75]. As an integral component of the super elongation complex (SEC), CDK9 kinase regulates the RNA polymerase II proximal pausing mechanism. The SEC is vital for cellular differentiation and development and can be misregulated in states of cellular transformation [76]. Suppressing aberrant SEC signalling in DMG cells was found to induce neuroglial differentiation with increases observed in GFAP, NGFR and NRN expression [75].

Targeting transcriptional elongation has been particularly efficacious in enhancer-driven malignancies where MYC aberrations are commonplace. Suppressing CDK7 and CDK9 activity has been known to substantially reduce MYC expression and downregulate MYC target genes in T cell acute lymphoblastic leukaemia, mixed-lineage leukaemia, neuroblastomas and small-cell lung cancers [77,78]. Interestingly however, SEC suppression exerts uniform anti-proliferative effects across DMG tumour subtypes irrespective of MYC expression [75]. However, translating these inhibitors into the clinic for DMGs may be challenging. Despite global changes in H3K27 posttranslational modifications transcriptomic changes are relatively restricted in DMG tumours [8]. Furthermore, BBB penetrance remains a significant hurdle for current transcription elongation inhibitors [75,79]. Precise understanding into the nature of off-target effects and effective drug design is critical for validating transcription elongation inhibition as a potential therapy for DMGs [77,78,80].

## 3. Targeting Cell Metabolism in DMGs

Targeting DMG metabolism is evolving as a potential treatment strategy, spurred on by the discovery of distinct links between metabolic reprogramming, mitochondrial dysfunction, and heightened levels of oxidative stress in DMG tumours, with significant advances also made in recent years in the understanding of the interplay between the DMG epigenome and metabolism. For example, glycolytic enzymes such as pyruvate kinase M2 (PKM2) have been implicated in H3K9 acetylation, causing a chromatin open state at *CCND1* (cyclin D1) and c-*Myc* loci, consistent with their subsequent activation [81]. PKM2 catalyses the rate-limiting step of glycolysis, shunting glucose metabolism away from oxidative phosphorylation towards anaerobic glycolysis and lactate production in tumour cells, a key feature during cancer development and progression. The depletion of PKM2 with regulatory microRNAs (miRNAs), long non-coding RNAs (lncRNAs) and circular RNAs (circRNAs) has yielded promising results in in vitro models of GBMs [82]. PKM2 expression, but not activity, is regulated in a grade-specific manner in gliomas, but changes in both PKM activity and PKM2 expression contribute to the growth of GBMs. The knockdown of PKM1/2 activated AMP-activated protein kinase (AMPK1) and suppressed viability in lung carcinoma cell lines [83]. AMPK1 plays an important role in maintaining H3K27 methylation deficiency. AMPK1 has been shown to directly target EZH2, disrupting the EZH2-dependent methylation of H3K27 and consequently PRC2 activity [13,84]. Furthermore, studies have shown that AMPK1 has a role in maintaining H3K27 methylation deficiency. AMPK1 has been shown to directly target EZH2, disrupting the EZH2-dependent methylation of H3K27 and consequently PRC2 activity [13,84]. Indeed, preclinical studies have demonstrated AMPK1 induction to suppress glycolysis through mTOR signalling, which was able to decrease tumour burden in DMG orthografts [23]. Additionally, pyruvate dehydrogenase kinase (PDK1) inhibition reverses the conversion of pyruvate into acetyl-CoA, uncoupling glycolysis from oxidative phosphorylation. It has been shown that simultaneous AMPK1 activation and PDK1 suppression is able to repress the glycolytic phenotype and enhance DIPG radiosensitisation within in vitro and DIPG orthograft models [85]. 

In a recent study, Chung et al. discovered that DIPG cells had enhanced glycolysis and tricarboxylic acid (TCA) cycle metabolism, resulting in elevated levels of α-ketoglutarate (α-KG). Targeting the metabolic enzymes that produce α-KG with the glutamine antagonist JHU-83 (oral gavage 20 mg/kg) and an IDH1 inhibitor (IP 10 mg/kg) increased survival in two orthotopic human DIPG PDX models [22]. α-KG has been well-documented as an important co-factor of JMJD3. Mechanistically, JMJD3 metabolizes α-KG to succinate, while demethylating H3K27me3. Inbuilt redundancies through the activity of hexokinase, WT IDH1 and glutamate dehydrogenase (GDH), which are heterogeneously expressed in both H3.3K27 and H3.1K27 DMG cells, may also serve as potential targets for therapeutic development. This work highlights the intimate relationship between metabolic and epigenetic mechanisms, indicating that disrupting metabolic pathways is an innovative strategy to deplete the essential co-factors required for maintaining H3K27 hypomethylation in DMGs.

Likewise, polyamines have been investigated in aggressive cancers as they play a pivotal role in multiple cellular processes and facilitate rapid cell proliferation. The intracellular concentration of polyamine is tightly regulated through biosynthetic and catabolic pathways as well as the uptake of polyamines from the microenvironment [86]. It has recently been shown that the polyamine pathway is not only upregulated within the setting of DMGs, but that combination treatment with the polyamine synthesis inhibitor difluoromethylornithine (DFMO) coupled with the polyamine transport inhibitor AMXT 1501 leads to a significant depletion of polyamine levels, resulting in a reduction in cell proliferation, clonogenic potential and cell migration, concurrent with the induction of apoptosis in DIPG neurosphere cultures. Furthermore, the combination of DFMO with AMXT 1501 significantly enhanced the survival of three orthotopic models of DMGs, further combining effectively with irradiation [24], with a treatment regime of DFMO and AMXT 1501 in combination currently being tested in an adult phase I clinical trial (NCT03536728) and a phase I trial in pediatric DMGs currently in development.

## 4. Targeting Cellular Signalling Pathways in DMGs

Since DMG tumours exhibit irregular activation of growth-factor-receptor-mediated signal transduction pathways, utilising drugs that target these pathways is a logical tactic. In vitro and in vivo studies have verified the efficacy of tyrosine kinase inhibitors, such as dasatinib, in diminishing tumour proliferation and inhibiting *PDGFRA* activity [87]. Phase I studies of PDGFR pathway inhibition by imatinib [88] and dasatinib [89], VEGFR2 inhibition by vandetinib [90] and EGFR inhibition by gefitinib [91] as well as erlotinib [92] revealed the safety of using these drugs in children and provided doses for future phase II studies. However, these inhibitors failed to show a clinical benefit when tested in further trials. Phase II trials for dasatinib (NCT00423735) and imatinib did not show clinically meaningful anti-tumour activity against recurrent adult GBMs [93,94]. The BIOMEDE trial was a clinically adaptive trial where DMG tumours from patients in Australia, Europe and the UK were specifically tested against three approved inhibitors targeting EGFR (erlotinib), mTOR (everolimus) or PDGFR (dastanib) for efficacy and in combination with standard radiotherapy followed by maintenance therapy (NCT02233049). However, there was no benefit observed, as measured by overall survival, with any of the drugs and this study was discontinued due to toxicity in 15%, 2% and 13% of patients, respectively [95].

Growth factor receptors are receptor tyrosine kinases (RTKs), and their downstream activity is facilitated through the activation of the specific RTK. RTK-dependent mitogenic activation plays a vital role in proliferation, invasiveness, cell survival and chemo- as well as radioresistance in a variety of cancers, including DMGs (Figure 1). DMG tumours regularly display focal amplifications in PDGFRA and EGFR accompanied by amplifications in KIT, KDR, EGFR and MET [96,97,98]. In vitro RTK suppression yielded promising results, although multikinase inhibition was more efficacious than specific-kinase inhibition, with only multi-kinase inhibitors such as dasatinib and crizotinib having thus far been shown to be effective [16,99]. Despite these positive results, both single- and multitargeted RTK inhibitors have failed to achieve relevant antitumor effects in vivo, ultimately failing to confer a survival benefit in phase I and II clinical trials relevant to DMGs [90,100]. The reasons behind the failure to target RTKs has been linked to the in-built redundancies in growth signalling pathways, tumoral heterogeneity, multidrug efflux transporters and the lack of specific and BBB-penetrant inhibitors [97,101]. Indeed, combination therapy regimes using dasatinib combined with cabozantinib, targeting hyperactive c-Met expression, demonstrated synergism against cultured DIPG neurospheres [87]. It is evident that there is still promise in the use of RTK inhibitors in treating paediatric malignancies with hyperactive RTK signalling. For example, the NCT03352427 study is investigating combination strategies that combine dasatinib with the mTOR inhibitor everolimus in DMG patients. 

The PI3K/AKT/mTOR intracellular signalling pathway is important in regulating the cell cycle, with mTOR activity, a common target for cancer therapeutics, being frequently upregulated across multiple cancer types. Growth factors, neurotransmitters and hormones are all able to activate the mTOR pathway through their specific RTKs and G-protein-coupled receptors (GPCRs) [102]. The first identified mTOR inhibitor, rapamycin, acts by binding to FK506-binding protein (FKBP) 12, forming a complex which then binds to mTOR, resulting in the downstream inhibition of the mTORC1 pathway [103]. Several analogues have since been developed due to the poor water solubility and pharmacokinetics of rapamycin. Two of these therapeutics, temsirolimus and everolimus, are currently available in the clinic, with temsirolimus approved for use in advanced renal cell carcinoma and everolimus for a range of non-CNS tumours in addition to subependymal giant-cell astrocytomas, which are benign tumours largely associated with tuberous sclerosis [104,105]. 

The PI3K/AKT/mTOR pathway has been identified as a promising target for therapeutics for DMGs due to its frequent dysregulation. Up to 50% of DMGs harbour mutations in genes upstream of mTOR, most prominently *PDGFRA*, *MET* and *IGFR1*, but also *EGFR*, *ERBB4*, *HGF*, *IGF2*, *KRAS*, *NF1*, *AKT1*, *AKT3*, *PTEN* and *PIK3CA*, leading to pathway overactivity [96,106,107]. PI3Ks are a family of intracellular signal transducers, consisting of three subunits: p85 regulatory subunit, p55 regulatory subunit and p110 catalytic subunit [108]. PI3Ks transmit signals from activated G-protein-coupled receptors and receptor tyrosine kinases, activating the downstream AKT pathway and subsequently mTOR signalling. mTOR is a serine/threonine protein kinase composed of two distinct complexes which differ in components, substrate specificity and downstream regulation: mTORC1, composed of mTOR, raptor, GβL and deptor, and mTORC2, composed of mTOR, rictor, GβL, PRR5, deptor and SIN1 [109,110]. mTORC1 exerts its downstream actions primarily through the phosphorylation of two proteins: p70S6 kinase 1 (S6K1) and eIF4E binding protein (4EBP). 4EBP and S6K1 are independent of one another, exerting their action on downstream substrates through distinctive approaches [111]. mTORC1 ultimately is involved in regulating the balance between anabolism and catabolism, promoting cell growth and glucose metabolism whilst suppressing autophagy, lysosomal biogenesis and proteasome activation [111,112,113,114,115,116,117]. mTORC2 acts via multiple mechanisms, including serine/threonine phosphorylation, the subsequent activation of AKT and the phosphorylation of members of the AGC family of protein kinases (PK), most notably from the PKA, PKB, PKC and SGK families [111,118,119]. mTORC2 activation results in increased cell migration and cytoskeletal rearrangement, decreased apoptosis and changes in glucose metabolism as well as ion transport [111].

While PI3K/AKT/mTOR inhibitors provide an attractive option for therapy, they are likely to be most effective when administered in combination with a second therapeutic due to the traditionally poor response of DMGs to single therapies (Figure 1). Indeed, temsirolimus has shown significantly increased efficacy against DMGs in vitro when combined with the mitochondrial protein adenine nucleotide translocase (ANT) inhibitor PENAO (4-(N-(Spenicillaminylacetyl)-amino)phenylarsonous acid), exhibiting increased apoptosis, decreased PI3K and mTOR signalling in addition to disrupted mitochondrial integrity in comparison to single-agent-treated cells. Unfortunately, these same promising results could not be recapitulated in vivo [23]. Similarly, PI3K/AKT and MEK/ERK inhibitors have been identified as a promising potential combination therapeutic for DMGs, with combination treatment resulting in a synergistic decrease in cell viability and an increase in apoptosis in comparison to single therapeutics in vitro [120,121]. Furthermore, PI3K/AKT inhibition with ZSTK474 and MEK/ERK inhibition with trametinib in combination reduced tumor burden compared to either agent alone; furthermore, median survival was significantly extended from 35 days in the vehicle controls to 47 days in mice treated with the combination therapy [121]. Combination targeting of PI3K/AKT/mTOR and HDAC has also shown efficacy in DMGs, with the dual PI3K/HDAC inhibitor CUDC-907 having been shown to decrease PI3K signalling and inhibit HDAC function in DMG and HGG cells in vitro. CUDC-907 was also identified as a radiosensitiser, with dual application of CUDC-907 and irradiation treatment resulting in a synergistic decrease in cell viability in vitro and a synergistic increase in animal survival [122]. A phase I clinical trial for CUDC-907 is currently underway for patients with DIPG (NCT03893487). Dual-targeting of the mTOR and CDK4/6 pathways has also been identified as a potential therapeutic option for DMGs [123], as discussed further below.

It has been recognised that mTORC1 inhibition can result in the detrimental upregulation of mTORC2, associated with increased tumor cell proliferation, migration and metabolism [111,123]. Subsequently, therapeutics capable of simultaneously targeting both mTORC complexes are considered advantageous. The dual mTORC1/2 inhibitor AZD2014 has shown significant efficacy against DMGs in vitro, being more efficacious than the mTORC1 inhibitor everolimus [124]. Another mTORC1/2 inhibitor, sapanisertib (TAK228), successfully suppressed DMG cell growth and invasion in vitro, and increased the survival of an orthotopic DMG model [125]. The addition of irradiation has been shown to further improve the response of dual mTORC1/2 inhibitors against DMGs, whilst the dual mTORC1/2 inhibitor GSK-458 was found to target DMG cells with stem-like cell qualities both in vitro and in vivo [124,125,126].

To date, two drugs targeting the PI3K/AKT/mTOR pathway have published results from DMG clinical trials: perifosine and temsirolimus [127,128]. As a monotherapy, the AKT inhibitor perifosine was found to be well-tolerated in children with DIPGs and other paediatric CNS and solid tumours at all tested doses (between 25 mg/m^2^/day and 125 mg/m^2^/day). Unfortunately, perifosine did not show significant efficacy in this trial, with the highest perifosine dose resulting in progressive disease in five of the six treated patients, across a range of pediatric CNS and solid tumours. Furthermore, two of the three DMG patients enrolled across the study exhibited progressive disease [127]. The combination therapy of perifosine and temsirolimus was arguably more successful, with all five DMG patients treated exhibiting stable disease at the first evaluation. However, across the study, which included patients with a range of recurrent or refractory solid brain tumours, including DIPGs, no significant responses were observed to the treatment [128]. The potential for therapeutically targeting the PI3K/AKT/mTOR signalling pathway in DMGs is evident, with several clinical trials currently underway. These include a monotherapy trial for paxalisib, a PI3K inhibitor (NCT03696355), a dual combination therapy trial of paxalisib and ONC201, a dopamine receptor D2 antagonist (NCT05009992), and temsirolimus combined with the HDAC inhibitor vorinostat, together with irradiation (NCT02420613). 

The activation of the PI3K/AKT/mTOR pathway is able to modulate TGFβ signalling in cancer [129]. The TGFβ superfamily is composed primarily of two distinct pathways involving TGFβ- and BMP-mediated signalling. The TGFβ pathway has known tumour suppressor properties, inhibiting cell growth and promoting apoptosis. The identification of mutations in components of the TGFβ pathway, such as *TGFBR2*, *SMAD2*, *SMAD3*, *SMAD4* and *ENG*, demonstrate the tumour suppressor role this pathway can play [130,131,132,133]. However, paradoxically, it can also act as a tumour promotor, enhancing tumour growth, invasion and metastasis [134]. TGFβ is especially associated with the epithelial-to-mesenchymal transition (EMT), a process associated with invasion, migration, metastasis and drug resistance [135,136]. BMP has a similarly paradoxical role in tumour suppression. Several BMP ligands are upregulated in a range of cancers, with BMPs able to stimulate tumour migration and invasion [137,138,139,140]. However, there is also evidence that BMP plays a suppressive role in tumorigenesis through several SMAD-independent pathways [141,142,143,144].

Both the TGFβ and BMP pathways signal through serine/threonine kinase receptors, resulting in downstream signalling through SMADs. Two distinct SMAD pathways are upregulated by the TGFβ superfamily: TGFβ ligands activate SMAD2/3 signalling whilst BMP ligands activate SMAD1/5/8 signalling [145]. These two distinct pathways result in upregulation in a wide range of genes involved in cell fate determination, cell cycle arrest, apoptosis and actin rearrangements [145]. The BMP pathway in particular is closely associated with DMGs. Mutations in *ACVR1*, which encodes a BMP type I receptor, activin-receptor-like kinase 2 (ALK2), are present up to 30% of DIPG patients [96,107,146,147]. ALK2 binds both BMP ligands, which activate the SMAD1/5/8 pathway, and activin A, which activates the SMAD2/3 pathway. *ACVR1* mutations result in hyperactivity in response to BMP ligands and constitutive BMP signalling, independent of ligand binding [148,149,150]. Furthermore, activin A promotes SMAD1/5/8 signalling in *ACVR1* mutated cells at even low ligand concentrations as a result of activin A forming ACVR2A/B–ACVR1R206H complexes [151,152,153,154]. *ACVR1* mutations in DMGs are associated with upregulated SMAD1/5/8 signalling and increases in cell proliferation, indicating an oncogenic role for the BMP pathway in this context [107,151,155]. Interestingly, *ACVR1* mutations appear to be unique to DMG tumors and are not present in other cancers; however, they are closely associated with the genetic condition fibrodysplasia ossificans progressive (FOP). FOP is a rare skeletal malformation disorder which results in aberrant, episodic heterotopic ossification (HO) of muscles, tendons and ligaments [156]. The progressive HO causes a range of skeletal pathologies, ultimately resulting in early death [157]. FOP is caused by mutations in *ACVR1*, primarily at residue 206 (R206H), which results in the classical, most severe phenotype [158]. Less common mutations are at the 258, 328 and 356 residues, resulting in a milder FOP phenotype [159,160,161]. Interestingly, the same mutations are observed in DMGs, with the R206H mutation shown to exhibit the most potent tumor phenotype. However, patients with FOP are at no greater risk of DMGs (or any other cancers), indicating that *ACVR1* mutations alone are not sufficient for tumorigenesis [146,147,151,155]. Previous work has identified a strong association between *ACVR1* mutation and the H3.1K27M mutation [96,107,146,147,151]. In addition, *ACVR1*-mutant DMGs exhibit a significant increase in mutations in PI3K-pathway-associated genes, whilst alterations to the TP53 pathway were less common [107,146,151]. 

The first small-molecule inhibitor of ALK2 that was identified was dorsomorphin, a compound capable of “dorsalising” zebrafish embryos and inhibiting BMP signalling [162]. Subsequently, further inhibitors based on the dorsomorphin pyrazolo[1,5-a]pyrimidine scaffold have been developed for use in both FOP and DMGs. A recent study examined eleven potential ALK2 inhibitors against DMGs, including compounds related to the dorsomorphin scaffold, pyridine compounds and drugs with reported anti-ALK2 activity. Nine compounds displayed efficacy against both *ACVR1*-mutant and *ACVR1*-wild-type DIPG cells, with very little selectivity for mutational status evident. In contrast, the two compounds tested in vivo, the pyrazolo[1,5-a]pyrimidine compound, LDN-193189, and the pyridine compound, LDN-214117, were only able to extend survival in an *ACVR1* (R206H)-mutant PDX model, with no improvement in survival seen in the *ACVR1*-wild-type PDX. However, it should be noted that whilst statistically significant the increase in survival was only modest, with both drugs increasing survival by 15 days [151]. The in vitro efficacy of LDN-193189 in DMGs has been observed by others, with some evidence of improved efficiency in *ACVR1*-mutant cells above those of the wild type; in contrast, another study into LDN-214117 found no significant effect on DMGs in vitro [107,153,155]. However, several analogues of LDN-214117 have been developed which exhibit a profile with increased potency, selectivity and BBB permeability. Preliminary studies have shown that they are efficacious against DMG cell lines, with some selectivity for *ACVR1*-mutant cultures [163]. A further analogue, M4K2127, has been identified as highly BBB-penetrable, including into the pons region of the brainstem [164]. Another ALK2 inhibitor, LDN-212854, a pyrazolo[1,5-a]pyrimidine compound, has also shown efficacy for DMGs, demonstrating sensitivity for *ACVR1*-mutant human DMG cell cultures in vitro. Furthermore, LDN-212854 was found to improve survival in a Nestin tv-a; p53fl/fl mouse model expressing *ACVR1* (R206H), H3.1K27M, PDGFA and Cre. However, similar to LDN-214117 this increase was only modest, with an eight-day increase in survival [155]. This work into ALK2 inhibitors is still at an early stage in DMGs. The preliminary results show that this area could be promising for developing new therapeutics; however, due to the varied results and few in vivo studies evident to date, more work is required in this field.

Several analogues of the currently used ALK2 inhibitors have been developed in attempts to increase and improve their potency, selectivity, BBB permeability and pharmacokinetic properties, although to date not all have been tested in DMG cells or PDX models. A modified structure of LDN-214117, known as M4K2149, has been developed and identified as a more potent inhibitor. M4K2149 analogues have also been developed with increased selectivity for ALK2, increased permeability and improved pharmacokinetic properties [165]. Analogues based on the pyridine compound K02288 have shown improved potency with increased cytotoxicity evident against HEPG2 cells derived from hepatocellular carcinoma. Interestingly, analogue cytotoxicity did not correlate with BMP signalling inhibition capacity [166]. Another area of focus has been inhibitors with an imidazo[1,2-a]pyridine scaffold, with a range of inhibitors identified with improved potency and PK properties [167]. It will be of interest to determine if these analogues are effective in DMG in vitro and in vivo models in the future.

Some surprising inhibitors of ACVR1/ALK2 have been identified. Binding between a MEK1/2 inhibitor, E6201 and ALK2 has identified E6201 as a potential BMP pathway inhibitor. This hypothesis was supported by its ability to dose-dependently inhibit BMP signalling in vitro and prolong DMG PDX survival in vivo [168]. OKlahoma Nitrone-007 (OKN-007) is an anti-cancer agent which acts through multiple mechanisms, including as an anti-angiogenic, GLUT-1 inhibitor and SULF2 enzyme activity inhibitor [169,170]. However, OKN-007 was also found to significantly reduce the expression of ALK2 in a DMG PDX model whilst also reducing tumor growth and increasing apoptosis, providing a promising potential therapeutic [171]. Due to the prevalence of the *ACVR1*-activating mutations in DMGs the BMP family has been the primary target of focus for DMG therapeutics, with little investigation into the TGFβ pathway. However, a recent study identified that the TβRI inhibitors EW7197 (vactosertib), LY3200882 and LY2157299 (galunisertib) were all effective in reducing DMG cell viability, with a further synergistic response seen with the addition of the HDAC inhibitor GSK-J4. Interestingly, these effects were most prominent in *ACVR1*-wild-type lines, with no response seen in the *ACVR1*-G328V-mutant DIPG line. It has been suggested that the constitutively activating *ACVR1* G328V mutation may suppress the TβRI pathway, thereby reducing the inhibitory efficacy of therapeutics [172]. Therefore, TβRI inhibitors may be a potential therapeutic option for *AVCR1*-wild-type DIPGs. Unfortunately, the effects of these drugs on downstream signalling in DIPG cells was not examined; this would be valuable due to the complex role of TGFβ signalling in cancer. The authors of this work suggest that the suppression of TβRI by the constitutively activating *ACVR1* mutation may contribute to the longer survival seen in DMG patients with *ACVR1* mutations, as it has been previously proposed that the increased kinase activity of mutant *ACVR1* may be involved in this phenomenon [107,146,172].

The ubiquitin–proteasome system (UPS), through the activity of several E3 ubiquitin ligases, plays a crucial role in the recognition and degradation of TGFβ family receptors, including SMAD components and their interacted proteins, to regulate TGFβ family signalling [173]. Additionally, the UPS pathway plays a critical role in facilitating neurogenesis and the growth of cerebellar granule cell precursors during brain development [174,175], with deregulation in the UPS being a well-known hallmark in a variety of CNS malignancies, including medulloblastoma and adult GBMs [176,177]. Ubiquitination by ligases, such as E3 ubiquitin ligase, play an integral role in the proteostasis of oncogenes and tumour suppressors, and subsequently regulate proliferation, DNA repair and apoptosis [174,178,179]. Furthermore, aberrations in E3 ligases dysregulate Shh and Wnt signalling, both of which result in medulloblastoma progression [177,180]. UPS suppression in medulloblastoma cell lines causes the activation of cell cycle checkpoints, with increases in the levels of cyclin-dependent kinase inhibitors p21 and p16 [181]. BBB-penetrant proteasomal inhibitors such as marizomib have demonstrated potent in vivo antitumor effects in orthotopic xenograft models of human GBMs and DIPGs [176,182]. A phase III study is now open for patients newly diagnosed with GBMs to study marizomib’s impact on overall survival (NCT03345095), with paediatric trials examining the use of proteasome inhibitors as a monotherapy versus a combination therapy for CNS tumours (NCT01132911 and NCT00994500) still ongoing.

Similarly, there is evidence that aberrations in the UPS, through E3 ligase dysregulation [183], extend to the Notch signalling pathway, itself also involved in cell proliferation, differentiation and survival, as well as being one of the most commonly activated signalling pathways in cancer [184], including DMGs [19]. Notch signalling is essential for maintaining stem-cell-ness, with the presence of quiescent stem-like cells in DMG tumours driving resistance to standard therapies and radiation [185,186,187]. In light of this, Notch inhibition with the γ-secretase inhibitor MRK003 has been shown to enhance radiotherapy-induced apoptosis in H3-K27M DIPG cells [19], with a co-dependency shown between H3K27me hypomethylation and Notch-mediated stemness in DIPGs in addition to the inhibition of key Notch pathway effectors ASCL1 and RBPJ demonstrating significant anti-tumour effects [188].

Studies have identified a potential role for cannabinoids in the regulation of the mTOR pathway, with cannabinoid-dependent increases in ceramide levels associated with the sustained activation of the Raf-1/MEK/ERK signalling cascade and the subsequent production of damaging cellular ROS and ER stress [189,190], culminating in the inhibition of Akt/mTORC1, triggering cell cycle arrest, autophagy and apoptosis [191]. The cannabinoids Δ9-tetrahydrocannabinol (THC) and cannabidiol (CBD) have been shown to possess some efficacy as potential therapeutic agents against adult brain tumours, particularly GBM (reviewed in [192]), acting via the G-protein-coupled cannabinoid receptors type 1 and 2 (CB1R and CB2R [193]). Compared to normal brain tissues, both low- and high-grade human gliomas are known to have increased CB2R expression on tumour cells, invading microglia/macrophages and endothelial cells of the tumour blood vessels [194,195,196]. However, the expression or abundance of CB1R or CB2R and their relevance in DMG biology is unknown [197], with the effects of cannabinoids within the pediatric setting, generally, remaining relatively understudied (reviewed in [198]). From the limited available data, Andradas et al. have recently demonstrated that THC and CBD exert cytotoxic activity against a range of paediatric medulloblastoma and ependymoma cell lines [193]. Here, it was determined that treatment with THC and CBD had variable effects on ROS production and the activation of MAPK or mTOR signalling in vitro, with no discernible benefit shown by either of the compounds within a pediatric medulloblastoma mouse xenograft model, alone or in combination with cyclophosphamide, one of the major chemotherapeutics used in clinical treatment. Taken together, the lack of therapeutic efficacy thus far demonstrated coupled with potential toxicity concerns [199] indicates that more research and beneficial evidence are required for cannabinoids to be a viable therapeutic option in DMGs.

## 5. Targeting the Cell Cycle and DNA Repair Mechanisms in DMGs

Aberrant cell signalling, leading to the constitutive initiation of the cell cycle and increased proliferation, is a hallmark of many cancers (reviewed recently in Matthews 2021) [200], including DMGs, presenting further potential therapeutic targets. For example, the cyclin-dependent kinase (CDK) 4/6 pathway is involved in the tight regulation of the cell cycle through mediation of the G1–S phase transition (Figure 2). Following CDK4/6 binding to cyclin D, the complex then phosphorylates the Rb protein. Phosphorylated Rb is unable to bind the E2F transcription factor, increasing its availability and thus downstream signalling [201]. Genes regulated by E2F include those involved in the cell cycle, DNA replication and mitosis [14]. The involvement of CDK4/6 in the cell cycle provides a key target for cancer therapeutics, with the pathway commonly targeted in breast cancer and three therapeutics currently approved for use in the clinic: palbociclib, ribociclib and abemaciclib. All three drugs target CDK4 and CDK6, with abemaciclib also being able to inhibit CDK9 [202]. The CDK4/6 pathway has been identified as a promising potential therapeutic target for DMGs due to its frequent dysregulation: approximately 30% of DMGs carry amplifications in genes involved in Rb signalling, including *CDK4*, *CDK6*, *CCND1*, *CCND2* and *CCND3* [106]. A study of more than 1000 cases of DMGs and pediatric HGGs identified co-segregation of the H3.3K27M mutation and G1–S phase dysregulation [203]. In contrast to other HGGs, deletions in *CDKN2A/B*, encoding the p16INK4A and p15INK4B tumour suppressors, are rare in DMGs; however, it has been shown that *CDKN2A* is epigenetically silenced due to the H3K27M mutation [106,203,204].

Whilst single therapeutic treatments have had little success in DMGs, CDK4/6 inhibitors have shown some efficacy as single agents pre-clinically. The CDK4/6 inhibitor palbociclib successfully reduced DMG cell proliferation in vitro and increased DMG PDX survival [205,206,207]. However, these effects have not been translated into the clinic; whilst both palbociclib and ribociclib have been well-tolerated by DIPG patients, phase I trials for both drugs have shown little survival benefit [208,209]. This was to some extent predicted by the preclinical data, which indicated that the in vivo doses of palbociclib required to achieve single-agent activity were not clinically achievable [210]. A phase I trial is currently underway for abemaciclib for patients with DMGs (NCT02644460). CDK4/6 inhibitors have shown the greatest efficacy when used in combination with secondary therapeutics. For example, a preclinical trial using PD-0332991 (PD), a CDK4/6 inhibitor, that induces cell cycle arrest both in vitro and in vivo in high-grade brainstem glioma cell lines enhanced survival when used in combination with radiotherpy in a genetically engineered, PDGF-B overexpressing, Ink4a-ARF and p53 deficient brainstem glioma mouse model [205]. Similarly, both irradiation and the EGFR inhibitor erlotinib were able to improve the efficacy of palbociclib in vivo [205,206]. A genome-wide analysis has previously identified 21% of DMGs with co-amplifications of the CDK4/6 and PI3K/AKT/mTOR pathway, providing an ideal dual target for combination therapy [106]. In addition, a study on glioblastomas found that mTOR inhibitors were able to reverse the increased mTOR signalling triggered by CDK4/6 inhibitors, with this compensatory relationship suggesting a promising potential combination therapy [211]. Subsequently, the dual inhibition of CDK4/6 and mTOR has shown promise for DMGs, with a combination treatment with palbociclib and the mTOR inhibitor temsirolimus synergistically reducing DMG cell proliferation in vitro [123]. Interestingly, palbociclib did not have any significant direct effects on mTOR signalling as a single agent in DMGs, in contrast to its actions on mTOR observed in glioblastomas [123,211]. Several phase I clinical trials investigating the efficacy of the dual combination of ribociclib and the mTOR inhibitor everolimus are currently underway for children with DMG and other HGGs (NCT03387020; NCT03355794). Determining if the synergistic effects observed in the lab can be translated into the clinic will be imperative.

It has been established that DMG tumours have aberrations in components of the DNA damage response (DDR) [212], which are coupled with the propensity of H3K27M mutations to lead to highly unstable genomes [96]. It is well-known that germline mutations in homologous recombination genes, for example *BRCA1*/2, predispose individuals to breast and ovarian cancer, and that these mutations are synthetic lethal with poly(ADP-ribose) polymerase (PARP) inhibition [213,214]. PARP binds to single-strand breaks to facilitate DNA repair; PARP inhibition results in an accumulation of double-strand breaks which cannot be repaired in homologous recombination (HR)-deficient cells, resulting in cell death [213]. PARP inhibitors are currently FDA-approved for *BRCA1*/2-mutant breast, prostate and ovarian cancer. It is now emerging that HR deficiency is apparent in a subset of DMG patients. Multiple members of the pathway undergo a loss of heterozygosity (LOH) or are deleted in DMGs, including *BRCA1, BRCA2, RAD50* and *RAD51L1* [215]. Furthermore, a recent integrated genomic analysis found that genetic alterations in DNA repair genes (including HR genes) were found in ~61% of DMGs [203]. These findings highlight HR-mediated DNA repair as a potential therapeutic target in DMGs. PARP1 was found to be expressed in DMG patient samples and cell lines, and DMG cells were sensitive to PARP inhibition with nirapirib [216]. Nirapirib reduced the rate of DNA repair and sensitised DMG cells to radiation [216]. Moreover, gain-of-function mutations in protein phosphatase 1D (*PPM1D*), a negative regulator of the DDR, were found in up to 29% of DMGs [217]. The inhibition of PPM1D sensitised *PPM1D*-mutant DMG cells to the PARP inhibitor olaparib, likely due to impaired HR-mediated repair [218]. Therefore, targeting DNA repair using PARP inhibitors may be a feasible strategy for HR-deficient DMG tumours.

Cell cycle progression is intricately linked to DNA repair and DDR activation, with DMG tumours relying on G1–S and G2–M checkpoint activation and arrest to halt cell cycle progression in order to allow time for DNA repair to take place [205,212]. Mutations in *TP53*, found in 42–77% of DMGs, have been hypothesized to be primary driver of radioresistance by relieving the TP53-mediated inhibitory control over the homologous recombination (HDR) activity, which primarily occurs during the G2 phase [219,220]. Moreover, *PPM1D* mutations are known to inactivate checkpoint kinases ataxia-telangiectasia mutated (ATM) and checkpoint kinases 1/2 (Chk1/2), impairing the initiation of the DDR after radiotherapy [218]. Given their role as a vital conduit, connecting the DDR to cell cycle machinery, targeting Chk1/Chk2 and ATM has been investigated as a therapeutic strategy [221,222]. Chk1 and Chk2 mediate TP53 activation and subsequently the facilitation of the DDR [220]. Chk1 inhibition with the Chk1-specific inhibitor prexasertib (LY2606368) exacerbated the anti-tumor effects of radiotherapy and suppressed RT-activated G1–S and G2–M checkpoint activation, allowing for DNA replication and mitosis to take place uninhibited. Moreover, radiosensitisation with prexasertib was particularly effective in *TP53*-mutant H3K27M cells but not in *TP53*-wild-type cells, where the latter remained arrested in G1 [220]. However, there are still challenges in translating the therapeutic potential of Chk1 inhibition into the clinic. A phase I clinical trial (NCT02808650) that investigated prexasertib in paediatric patients with recurrent or refractory solid and CNS tumours did not show any objective responses, indicating that it was well-tolerated [223].

Chk1/2 are known to target key players in G2/M progression: Wee1 and polo-like kinase 1 (PLK1), both of which are found to be overexpressed and have themselves been targets of investigation in recent years [212]. Wee1 is a serine/threonine checkpoint kinase that acts as a gatekeeper of cell cycle progression. Following DNA damage, Wee1 is activated by Chk1/2, triggering G2–M checkpoint activation. Wee1 phosphorylation and subsequent degradation by PLK1 relieves Wee1′s inhibitory control over cyclin-dependent kinase 1 (CDK1) at the G2/M checkpoint, enabling the CDK1/cyclin B complex to drive the G2–M transition [224]. Werbrouck et al. identified a synthetic lethal interaction between RT and the knockdown of Wee1 and PLK1 in TP53-mutant H3-K27M DMG cells [220]. Targeting Wee1 with the small-molecule inhibitor adavosertib (MK1775/AZD1775) attenuated radiotherapy-induced G2/M arrest, forced mitotic entry in DMG cells and exerted anti-tumor effects in DMG orthografts [225,226]. However, Wee1 is also thought to directly exert DNA damage irrespective of its effect on the cell cycle, although the underlying mechanisms are unclear [226]. Interestingly, both CDK1 and Wee1 are known to phosphorylate EZH2 and facilitate its ubiquitination [227,228], although this interplay between epigenetic regulation and cell cycle mechanisms is yet to be fully explored. The NCT01922076 trial is currently assessing the safety, toxicity and MTD of concurrent adavosertib treatment with radiotherapy in DMG patients.

PLK1 is a key regulator of the G2–M transition and mediates centrosome maturation, spindle assembly mitotic entry, the metaphase-to-anaphase transition and cytokinesis [229,230]. PLK1 exerts its effects on the G2–M transition via the activation of the M-phase inducer phosphatase 3 (CDC25C), which subsequently dephosphorylates and activates the CDK1/cyclin B complex [231]. In addition, PLK1 drives mitotic entry via Wee1 degradation, which relieves Wee1′s inhibitory control over CDK1 [232]. The forced mitotic entry caused by PLK1-mediated Wee1 degradation is thought to play an important role in cell recovery following DNA damage and G2 arrest [228]. However, functional TP53 is known to attenuate PLK1-inhibition-induced cytotoxicity by activating the DDR via ATM and ATR as well as permitting centrosome separation in colon carcinoma cell models [233]. As such, the extent to which TP53 modulates sensitivity to PLK1 inhibitors requires further investigation in DIPG cell line models.

The therapeutic impact of targeting PLK1 is relatively unknown in paediatric gliomas, with only one study describing the in vitro efficacy of direct PLK1 inhibition in DMGs [234]. Here, it was found that the targeting of PLK1 with the PLK1 selective inhibitor volasertib (BI 6727) exerted anti-tumour effects against a range of cultured DMG cell lines, with PLK1 inhibition leading to significant G2-to-M-phase cell cycle arrest and H3.1K27M-mutant DMG cell lines showing more sensitivity than their H3.3K27M counterparts. It was further shown that the inhibition of PLK1 with volasertib acted to sensitise DMG cells to the effects of irradiation [234], further highlighting PLK1as a therapeutic target beyond its role in cell cycle progression towards a further role in DNA damage repair mechanisms.

## 6. Activating the Immune Response as a Potential Therapeutic Option in DMGs

Examination of the differences between the immune profiles of both pediatric HGGs and DMGs has demonstrated that DMGs typically have higher leukocyte chemo-attractant expression, with immune cells constituting a large portion of the cells within the DMG tumour mass. However, the immune environment in DMGs is largely non-inflammatory, with a low adaptive immune response and decreased infiltration of natural killer (NK) cells (reviewed in [235]). Of the immune cells present, bone-marrow-derived macrophages (BMDMs), over microglia, are the predominant tumour-associated macrophages/microglia (TAM) subpopulation in DMGs [25]; it has been established that human DMG samples express high levels of the pan-macrophage markers CD11b, CD45 and CD68 concurrently with increased expression of the M2 marker CD163 [236], confirming that high expression of chemo-attractants results in the enhanced infiltration of, in particular, alternatively activated macrophages within this disease setting. Despite this knowledge, therapeutic targeting of non-tumour-infiltrating cell types, including TAMs, has only recently begun to be investigated.

Macrophages retain an inherent plasticity, capable of being polarised to either a classically activated, proinflammatory M1 or an alternatively activated M2 phenotype depending entirely on the inflammatory environment encountered (Figure 3). It is this very plasticity that makes macrophages a prime therapeutic target in many forms of cancer. Classically activated macrophages assume the M1 phenotype, which is characterised by an increased expression of the STAT1 signalling pathway and iNOS activation [237]. These cells are capable of stimulating an antitumor immune response through presenting antigens to adaptive immune cells, producing proinflammatory cytokines such as IL-1β, IL-6, IL-12 and TNFα in addition to chemotactic factors such as IL-8 and MCP-1 [238,239], as well as by phagocytosing tumour cells [240,241]. In comparison, alternatively activated M2 macrophages are characterised by the activation of the STAT3 pathway, resulting in the expression of the scavenger receptors CD163, CD204 and CD206 [242], as well as the production of immunosuppressive cytokines, including predominately Arg-1 [243], TGF-β [244] and IL-10 [245].

In support of the consensus that DMGs have been historically considered “immune cold” with respect to infiltrating leukocytes, Lin et al. have subsequently demonstrated that, compared to the situation in GBMs, there exists a low intrinsic inflammatory signature that contributes to the non-inflammatory phenotype of DMG TAMs [26]. Here, building on the earlier obtained data [236], pre-ranked gene-set enrichment analysis on DMG TAMs found that there were no significant increases in either M1- or M2-defined gene sets, with the authors concluding that DMG TAMs do not easily fit into a strict M1 or M2 classification [26]. These results mirror data collected from studies within the setting of gliomas [246] and GBMs [247]. Within these disease settings, the majority of TAMs, and likewise microglia, as well as macrophages isolated from normal brain regions could not be categorised into individual polarisation phenotypes. Indeed, comparative profiling of TAMs with matched controls, including circulating blood monocytes, non-polarised M0 and polarised M1 or M2 macrophages, indicate that macrophages that have infiltrated into GBM tumour tissue exhibit a continuum state more consistently resembling an undifferentiated M0 macrophage state [247], with similar results demonstrated in gliomas [246]. From these data, as well as those generated from a previous preliminary study [248], comprehensive analysis found that GBM TAMs also express comparatively lower levels of the M2 markers CD163, TGFβ and IL-10 compared to control macrophage populations specifically polarised to an M2 phenotype despite demonstrating increased activation of the STAT3 signalling pathway, a characteristic indicative of an M2 phenotype [247]. Additionally, the authors here also demonstrated that GBM TAMs express lower levels of the M1 markers IL-1β and IL-6 relative to control M1-polarised macrophages [247,248].

Taken together, these results suggest the complex nature of TAMs in general but overall confirm that within these disease settings there exists a low inflammatory state, with the propensity of TAMs to exhibit a primarily M0 basal phenotype. Indeed, authors of these studies concluded that the thorough elucidation and identification of the mechanisms and pathways associated with an M0 macrophage phenotype alignment provide the basis for the development of macrophage-targeted therapeutic strategies focussed on propelling TAMs from this evident M0 towards an M1 inflammatory phenotype [247], thereby promoting the T cell Th1 proinflammatory cytokine immune response, resulting in tumour suppression [249]. As it is known that there is minimal active T cell infiltrate within DMGs [26], promoting the immune response by targeting M1 macrophage phenotype activation mechanisms would result in greater tumour regression and clearance.

There are a number of potential DMG therapeutics that target diverse cellular processes including the cell cycle, metabolism and epigenetic mechanisms that have the concurrent ability to modulate the macrophage phenotypic switch towards an M1-dominant TAM population, required to assist in tumour clearance. For example, the mTOR signalling pathway plays a central role in the metabolic programming of TAMs [250]; downstream of mTOR activation the activation of STAT3 negatively regulates the macrophage-derived antitumor immune response [251] by conferring an M2 phenotype [242]. In addition to having a directly positive effect against DMG cells [124], targeting of macrophage mTOR activity with rapamycin has been shown to stimulate macrophages towards an M1 phenotype, contributing to an overall anti-tumour effect within the setting of GBMs [252]. Similarly, in gliomas, a gain-of-function mutation in STAT3 results in increased immunosuppression and heightened tumour invasion, whilst within the setting of DMGs specifically it is known that STAT3 is elevated. In confirmation of its role in DMGs, exposure to the JAK2-specific STAT3 inhibitor AG490 resulted in decreased DMG cell invasion, migration and viability, whilst sensitising DMG cells to radiation by interfering with DNA damage repair mechanisms [253]. Whilst these studies did not extend to in vivo analysis of DMG tumorgenicity in response to AG490 therapy, it is interesting to note that it is also known that the STAT3 inhibitor AG490 is able to abrogate M2 macrophage polarisation. For example, within the setting of multiple myeloma, exposure to AG490 drives macrophages from an alternatively activated state, previously invoked by being co-cultured with tumour cells, towards an inflammatory phenotype. This was evidenced by the decreased expression of M2 markers Arg-1, CD163 and CD206 that was concurrent with a significant elevation in TNFα expression, characteristic of an M1 phenotypic switch, demonstrating the dual role that STAT3 inhibition may play within this disease setting [254].

Likewise, PLK1 is a key regulator of the cell cycle that demonstrates heightened expression in DMGs, making it an attractive therapeutic target. Indeed, targeting of PLK1 with volasertib demonstrates anti-tumour effects against DMG cell lines in vitro, where it leads to significant cell cycle arrest and apoptosis [234]. Highly expressed PLK1 promotes TNFα-stimulated gene 6 (TSG6) signalling and enhances an invasive phenotype in lung cancer cells [255], with TSG6 activation preventing the macrophage expression of proinflammatory M1 markers such as iNOS, IL-6, TNFα and IL-1β while increasing the expression of anti-inflammatory M2 markers such as CD206, IL-4 and IL-10 [256]. These results suggest that targeting PLK1 may also invoke a favourable immune response capable of enhancing tumour clearance. However, it has also been shown within a THP-1 monocytic cell line model of macrophage function that PLK1 inhibition, mediated via the PLK1 inhibitor GW843682X, significantly decreased lipopolysaccharide (LPS)-induced TNFα mRNA expression in a dose-dependent manner. The authors here attributed this apparent decrease in the M1 macrophage response to LPS stimulation to the inhibition of the MAPK and NFKβ signalling pathways by the PLK1 inhibitor [257]. Whilst the role of PLK1 in DMGs within an in vivo setting is yet to be functionally determined, the non-inflammatory tumour microenvironment within this setting would support the absence of macrophage toll-like receptor 4 (TLR-4) engagement, required for LPS-induced signalling [258] within the macrophage cell model used above [257].

The consistently inactivated state of TAM TLR-4 signalling may itself present a viable therapeutic target in DMGs. As an example of how this may be effective, it has been demonstrated that TLR-4 activation, using the chemotherapeutic agent paclitaxel, was able to polarise immunosuppressed M2 macrophages towards an M1 state, with their inflammatory activation able to inhibit tumour progression in both breast cancer and melanoma models, with TAMs isolated from tumour-bearing mice treated with paclitaxel exhibiting an increase in M1 phenotype iNOS, IL-6 and IL-12 expression concurrent with decreased M2 phenotype expression of CD206 and Arg-1 [259]. Whilst paclitaxel has been reported to display limited efficacy when previously trialled as a radiosensitiser in DMGs [260], it may yet prove useful within this disease setting as a specific beneficial modulator of the macrophage inflammatory response.

Similarly, the targeting of polyamine synthesis and transport has been shown to be an effective therapeutic against DMG tumorgenicity whilst also being known to drive an M1 macrophage inflammatory state. Khan et al. [24] have shown that polyamine synthesis is upregulated in DMGs. When the rate-limiting enzyme in this process, ODC1, is targeted with DMFO treatment compensatory mechanisms are invoked that result in enhanced polyamine transport through upregulation in the cellular membrane transporter SLC3A2. Confirmation of the pivotal role the polyamine pathway plays in DMG tumorgenicity was subsequently ascertained, both in vitro and in vivo, through the dual-targeting of polyamine synthesis with a combined therapeutic regime using DFMO to target polyamine synthesis, coupled with the inhibition of polyamine transport using the compound AMXT 1501 [24]. Whilst the authors here did not seek to ascertain the inflammatory status within their mouse PDX DMG model following polyamine pathway targeting, a similar study within the setting of melanoma provides an informative indication. Alexander et al. [261] have targeted melanoma tumour growth in mice by treatment with DFMO combined with the polyamine transport inhibitor trimer PT1. In this study they found that this combination therapy concurrently decreased tumour growth whilst significantly increasing proinflammatory TNFα, IFNγ and MCP-1 cytokine and chemokine expression, concomitant with a significant reduction in the M2-polarised macrophage population [261]. These studies highlight that, although not routinely examined within the setting of DMGs, targeting pathways commonly expressed in tumours and infiltrating TAMs such as mTOR and STAT3 signalling or polyamine synthesis and transport may provide an additive effect that disrupts the cross-talk between immune and malignant cells, effectively reducing immunosuppression and promoting tumour sequestration and removal [262].

Epigenetic dysfunction resulting from H3K27M mutations are frequently observed in DMGs, with the targeting of epigenetic regulation and associated mechanisms offering viable treatment options (Table 1). For example, DMGs have been shown to be exquisitely sensitive to HDAC inhibition [16], with the specific role of HDACs in DMG epithelial-to-mesenchymal transition (EMT), which results in increased tumour invasiveness, being subsequently shown in [263]. In this latter study, treatment of DMG cultures with up to 200 nmol/L of the HDAC inhibitor panobinostat significantly decreased the protein expression of EMT transcription factors, including ZEB1 and SNAIL/SLUG. Furthermore, panobinostat exposure also led to a concomitant decrease in the expression of stem cell phenotype markers SOX2 and NESTIN, implicating a direct connection between the epigenetic regulation of both the mesenchymal and stem cell characteristics of DMGs [263]. SNAIL expression has been implicated in the modulation and secretion of cytokines that can influence the tumour immune infiltrate, with tumour-cell-specific SNAIL deletion within the setting of breast cancer causing a higher percent of TAMs to be polarised towards an M1 phenotype coupled with a decrease in the percent of M2 macrophages [264]. In support of this, the treatment of DMGs with panobinostat and the resultant histone hyperacetylation induces an interferon type I response. This results in the increased expression of interferon-stimulated genes, which can sensitise tumour cells to the innate immune system [42] by potentiating the activation of the M1 macrophages’ response [265,266] and enhancing effector T cell differentiation [267].

Similarly, the use of the HDAC inhibitor trichostatin-A (TSA) at 0.5 µM/kg in mouse models of melanoma or breast cancer was shown to be dependent on an active immune system, with treatment shown to be beneficial in C57BL/6 mice but have no effect in athymic nude mice. TAMs isolated from C57BL/6 mouse tumour models treated with TSA exhibited decreased expression of the M2 markers Arg-1 and CD206, with heightened expression of the M1 markers iNOS and IL-6 [268]. Furthermore, the authors here conclude that TAMs act as mediators of HDAC-inhibited immunomodulatory activity, directly leading to tumour suppression. In support of this, TAMs from TSA-treated mice were adoptively transferred to new tumour-bearing recipient mice. Compared to controls, inoculation with the TSA-modulated TAMs resulted in a significant decrease in tumour burden in the recipient mice, suggesting that HDAC inhibition epigenetically rewires macrophages into an M1 state, decreasing phenotypic plasticity and thereby stabilising their anti-tumour activity. This study went on to highlight the therapy-evasive nature of tumours, finding that TSA treatment, whilst effective at modulating TAM function, also increased programmed death ligand 1 (PD-L1) expression. PD-L1 expression is an immune evasion mechanism exploited by various malignancies and generally associated with poorer prognoses [269], including within DMGs where its expression is correlated with the extent of tumour-infiltrating lymphocytes [270,271]. In their study, Li et al. found that a dual-targeted approach was required in order to effectively combat tumour reoccurrence, combining TSA with anti-PD-L1 therapy to significantly enhance tumour reduction durability and prolong survival of tumour-bearing mice [268].

It is accepted that any potential DMG therapy regime would rely on a combination of treatment options tailored to the ongoing status of individual patients. The significance of the above study may therefore be in highlighting the potential therapeutic advantage of targeting epigenetic regulation in DMGs and the additive, cross-over effect such a strategy would have on the regulation of the immune response. It is known that the H3K27M mutation alters the methylation ability of EZH2 within the PRC2 complex, with this epigenetic mechanism having a profound impact on DMG/DIPG tumorgenicity. Whist directly targeting EZH2 in DIPG cells demonstrates limited efficacy, Keane and co-workers have recently demonstrated that EZH2 activity also regulates DIPG-induced microglia activation. Here, it was shown that the targeted siRNA knockdown of EZH2 resulted in a phenotypic switch of microglia towards an M1 state, indicated by the increased expression of iNOS [272]. This is consistent with the ability of EZH2 activation to repress the M1 macrophage inflammatory response, with the targeted inhibition of EZH2 by exposing macrophages to the EZH2 inhibitor UNC199 able to induce a dose-dependent increase in the LPS-induced expression of the M1 polarisation marker IL-6 [273]. These results suggest that EZH2 inhibition may exert its clinical benefit within the setting of DIPGs by targeting the tumour microenvironment rather than the tumour cells themselves, leading to a proinflammatory microglia and macrophage state that favours tumour clearance [272].

Taken together, these data suggest that beneficially modulating the immune response in parallel to, or combination with, DMG tumour targeting may work to enhance patient therapy, highlighting the need for a more holistic approach when evaluating potential treatment options for the disease. Additionally, there is a direct link between TAM phenotype status and subsequent T cell activation, which bears consequences for immunomodulatory therapy for brain tumours. Although the precise mechanisms in DMGs remain to be fully elucidated, it has been shown that GBM tumour cells recruit and polarise both macrophages and microglia to an M2 phenotype, resulting in the inhibition of T cell proliferation through the production of the immunosuppressive cytokines TGFβ and IL-10 [274], whilst also preventing the production of cytokines required to support activated tumour-specific CD8+ T, CD4+ Th1 and Th17 cells [247]. Mechanistically, GBM tumour cell kynurenine production activates the aryl hydrocarbon receptor (AHR) on TAMs, decreasing NFKβ signalling and thereby promoting an M2-like phenotype, leading to cytotoxic CD8+ T cell dysfunction [275]. Dysfunctional T cells are defined by the loss of effector function, including the loss of cytotoxicity and the decreased secretion of inflammatory cytokines such as TNFα and IFNγ [276] which, in turn, decrease M1 TAM polarisation, perpetuating the low inflammatory microenvironment characteristic of this disease. Further, the increased expression of Arg-1 by M2 macrophages is able to directly suppress T cell function within the microenvironment through the arginase-mediated depletion of arginine, which induces the down-regulation of the T cell CD3ζ chain [277]. It being the case that the activation domain of the CD3ζ chain is the major feature of the intracellular portion of the chimeric antigen receptor (CAR) of T cells, TAM overexpression of Arg-1 within the tumour microenvironment can be seen as a direct impediment for successful immunological targeting in DMGs using CAR T therapy.

In addition to targeting the upstream activation of the inflammatory response through modulation of the TAM phenotype switch, augmentation of T cell activation by employing CAR T therapy is another potential therapeutic avenue aimed at beneficially altering the DMG tumour microenvironment. In CAR T therapy, peripherally circulating T cells are primed against antigens by ex vivo co-culturing with antigen-loaded dendritic cells or, alternately, are modified with a CAR gene, arming these cytolytic T cells with a receptor that can recognize a surface protein on tumour cells [276]. These cells are then expanded in culture before being infused into patients as an adoptive T cell transfer [278]. Using CAR T cells to deliver antibodies in this manner overcomes the inability of the antibodies alone to cross the BBB.

The primary hurdle to successful CAR T cell therapy is the engraftment of cells, involving both the sufficient migration of the transferred cells to within the tumour microenvironment and the sustained activation of their anti-tumour responses [276]. One approach to improve engraftment is the pre-conditioning of patients with chemotherapy to induce lymphopenia, with this approach currently in the recruitment stage of a phase I clinical trial within the setting of DMGs (NCT03396575). Another barrier to successful T cell therapy in DMGs is the lack of a sustained level of potency in the transferred T cells. The decreased efficacy of transferred T cells can be overcome through the concurrent therapeutic use of antibodies targeting immune checkpoints. Immune checkpoint molecules are frequently overexpressed during tumour development as a mechanism by which tumour cells are able to subvert the immune system response. A range of these molecules are known to be expressed within the tumour environment, including PD-L1/PD-1, cytotoxic T-lymphocyte antigen 4 (CTLA-4) [279], lymphocyte activation gene 3 (LAG-3) [280], B7-H3 [278] and indoleamine 2,3-dioxygenase (IDO) [281], although relatively few of them have been evaluated within the setting of DMGs as potential therapeutic targets [235]. For example, the disruption of the PD-L1/PD-1 immune inhibitory axis is known to be a mechanism by which other tumours evade the immune system [282]. In support of anti-PD-L1 therapy significantly enhancing tumour reduction and prolonging survival of tumour-bearing mice by augmenting the HDAC inhibitor modulation of the M1 TAM response detailed above [268], PD-1 blockage has also been shown to be effective at aiding the expansion and prolonged potency of CAR T therapy [282]. Unfortunately, in a comprehensive screen of cell surface antigens present on patient-derived DMG cultures, Mount et al. ascertained that both PD-1 and CTLA-4 had relatively low expression [283], indicating that the further targeting of these molecules within this setting, in combination with CAR T therapy, may not be effective. In contrast, Majzner et al. found that cell surface expression of the checkpoint molecule B7-H3 is high on DMGs. The authors went on to show that CAR T cells directed at B7-H3 are capable of crossing the BBB and produce sufficient tumour-killing amounts of IFNγ, IL-2 and TNFα when cocultured with DMG cells [278]. It remains to be seen whether these successful preliminary studies can be recapitulated within an in vivo model of DMGs. In their screen for potential targets for CAR T cell immunotherapy in DMGs, Mount et al. identified disialoganglioside GD2 as being very highly expressed within this disease setting [283]. The upregulation of the GD2 antigen in brain tumours compared to a normal brain makes it an attractive target for immunotherapy [284,285]. Similar to the above study [278], the co-culturing of GD2-CAR T cells with DMG cells triggered the production of IFNγ and IL-2. Furthermore, the authors were able to demonstrate within a range of DMG PDX mouse models that GD2-CAR T cell therapy was accompanied by widespread inflammatory infiltrate to within the tumour microenvironment, effectively clearing tumours, evident as early as day 14 of treatment [283]. They did, however note that tumour clearance was not total, with the persistence of tumour cells that were negative for GD2 expression following GD2-CAR T cell treatment. Worryingly, the authors also noted substantial toxicity occurred in GD2-CAR-T-cell-treated mice during the period of maximal therapeutic effect, occurring from days 10-14. This outcome was attributed to a heightened inflammatory response in brain regions that were susceptible to increased intracranial pressure and lethal transtentorial herniation, such as the thalamus. These occurrences are unfortunately not isolated and are recognised as potential neurotoxic complications of CAR T cell therapy, particularly in combination with checkpoint inhibitors [283]. Although demonstrating potential, the risks of toxicity along with the current scarcity of information of any other potential DMG targets, including immune checkpoint expression, need to be addressed in order to accomplish the sustained level of potency required for effective CAR T therapy in DMGs.

## 7. Targeting of Neuronal Cell–DMG Interactions

DMGs are highly infiltrative tumours in the brain. At the histological level glioma cells have been observed to display a specific morphology that surrounds neurons, a hallmark now described as Scherer’s structures or perineuronal satellitosis [286,287] (Figure 4A). This feature is not unique to neoplastic cells but is also observed in healthy tissues between glial cells and neurons [288,289]. The interaction between oligodendrocyte precursor cells (OPCs) and neurons is essential for the production of myelin, which protects and supports axons during the propagation of action potentials [290]. Increased neuronal activity was found to have mitogenic effects on neural precursor cells (NPCs) and OPCs in juvenile and adult brains [291]. Although the exact cell of origin of DMGs is not understood, NPCs and OPCs have been reported to give rise to gliomagenesis [97,292]. Recent studies through a series of optogenetic experiments demonstrated that neuronal activity could stimulate the growth of HGG and DMG cells through the secretion of synaptic factors such as neuroligin-3 (NLG3). Although the exact receptor at which NLG3 acts upon tumour cells has not been identified, it has been shown that the subsequent activation of focal adhesion kinase (FAK) and downstream signalling through the PI3K/AKT pathway leads to glioma cell proliferation [49]. Subsequently, the same authors demonstrated that HGG/DIPG growth was decreased in *Nlg3* knockout animals. NLG3 release from neurons could be inhibited upon treatment with tetrodotoxin, a voltage-gated sodium channel blocker, indicating that NLG3 release is mediated upon neuronal activity. Furthermore, it was found that NLG3 could be cleaved by a disintegrin and metalloproteinase domain-containing protein 10 (ADAM10) (Figure 4B). Therapeutic inhibition of ADAM10 and ADAM10/17 with GI254023X and INCB7839, respectively, significantly reduced HGG and DMG growth in vivo, as observed by Xenogen imaging. Although treating animals with both inhibitors did not exhibit toxicity, long-term effects on neurofunction need to be investigated carefully [27]. INCB7839 is BBB-permeable and currently under investigation in HGGs, including DIPGs (NCT04295759). Although FAK inhibition could abrogate the effects of NLG3, its therapeutic capacity is limited due to low permeability in infiltrative tumours such as DMGs [293]; as yet, PI3K/AKT/mTOR pathway inhibitors have not been explored for their potential to attenuate synaptic communication within this setting.

Based on these earlier findings, Monje’s team hypothesised that DMGs might engage directly in neuron communication through synapse formation [294]. Single-cell transcriptomic analysis of DMG tumours indicated distinct subpopulations with features resembling astrocytes, oligodendrocytes and OPC-like cells, later demonstrating enrichment of synapse-related genes [97,294]. Electron microscopy and electrophysiology experiments indicated the presence of functional synapses between DMG cells and neurons. More specifically, whole-cell patch-clamp experiments measured two distinct electrophysiological responses in DMG cells upon neuronal stimulation: a rapid excitatory postsynaptic current (EPSC, <5 ms) and a rather prolonged (>1 s) current. EPSC was found to be mediated by calcium-permeable (GluA20 α-amino-3-hydroxy-5-methyl-4-isoxazolepropionic acid receptors (AMPARs) and subsequently confirmed to be inhibited by AMPAR antagonists [294] (Figure 4C). Furthermore, treating animals with the AMPAR inhibitor perampanel, an antiepileptic agent, resulted in a significant decrease in proliferative Ki67-positive cells in orthotopic animal models of DMGs. Perampanel has so far been investigated as an antiepileptic agent in brain tumour patients, with promising responses being observed; however, further research is needed to understand which patients may benefit and whether it has anti-glioma properties [295]. HGGs were recently reported to form neurite-like, microtube-connected multicellular structures as a mechanism to communicate and resist treatment [296]. These microtubes were connected through hexameric structures consisting of connexin 43 protein, forming gap channels [296]. Consistent with this notion, the inhibition of gap junctions with meclofenamate repressed the propagation of prolonged currents upon neuronal stimulation and reduced the proliferative index in treated orthotopic DMG animals [294] (Figure 4D). Although these studies collectively demonstrate that neuronal hyperactivity contributes to glioma growth, Yu and colleagues showed that specific PI3K mutations might also influence the neuronal microenvironment [297]. They found that animals with GBM tumours containing specific phosphatidylinositol-4,5-bisphosphate 3-kinase catalytic subunit alpha (*PIK3CA*) mutations (C420R and H1047R) exhibited more seizures, while transcriptomic analysis revealed higher expression levels of genes involved in the formation of synapses [297]. Specifically, in the presence of the C420R variant, further experiments implicated the expression of glypican 3 (GPC3) as the key factor for the increased neuronal hyperactivity. *Gpc3*-knockout glioma cells with *PIK3CA* C420R exhibited prolonged survival compared to wild-type *Gpc3*. Currently, GPC3 inhibition is limited to immunotherapy approaches with antibody-based and CAR T therapies under clinical investigation in adult hepatocellular carcinoma [298]. Given the presence of *PIK3CA* mutations in DMGs, it is still to be confirmed if these variants can initiate neuronal activity as observed for GBMs and whether it is mediated through glypicans. Astrocyte-derived glypicans have been shown to initiate functional synapse formation and increase the recruitment of AMPARs to the synapses [299]; however, this must be yet confirmed in the context of DMGs (Figure 4E). Another intriguing matter is that endogenous polyamines can modulate calcium-permeable AMPARs. Polyamines can enter the water-filled regions of the AMPARs and, with their positive charge, block the flow of small ions such as Na^+^ and Ca^2+^ [300]. NASPM, a synthetic polyamine, was found to play a partial neuroprotective role during ischemia in rats [301]. In addition, Venkatesh et al. showed that NASPM treatment inhibited the rapid EPSC but not the prolonged current in glioma cells [294]. As discussed previously, the metabolism of polyamine was found to be upregulated in DMG tumours, and dual inhibition of synthesis and uptake demonstrated significant extension of survival in orthotopic models of DMGs [24]. Although polyamines are known to play an important role in the proliferation of tumour cells, their potential role in modulating neuronal hyperactivity has not been investigated in DMGs. It is possible that at high levels polyamines may play a protective role at the local level where calcium-permeable AMPARs mediate the propagation of action potentials (Figure 4C).

It is currently unknown if other neurotransmitter receptors and molecules promote DMG proliferation through synapse formation. In other aggressive brain tumours, such as GBMs, glutamatergic and dopamine receptors have been implicated in their growth and migration [302] as well as having been recently reviewed in [303,304] (Figure 4C). Downstream mediators of neurotransmitter pathways are starting to be realised as potential pharmacological targets. Based on phosphoproteomic analysis performed across a range of paediatric tumours, calmodulin-dependent protein kinase II alpha (CAMKIIα) was reported to be elevated in the specific “neuronal class” of HGGs, which also exhibited higher expression of genes involved in glutamate neurotransmission and synaptogenesis [297]. CAMKIIα has been shown to play a significant role in synaptic transmission; thus, in the context of brainstem gliomas, it could potentially be implicated in propagating neuronal activity [305,306] (Figure 4C). Currently, highly selective and potent CAMKII inhibitors are not available; however, the development of new inhibitors remains an active area of research for neurological and cardiovascular diseases [307]. Other targets along the CAMKII pathway, such as calmodulin, have not been extensively explored. A few studies suggest antiproliferative, anticancer effects when targeted with antipsychotic agents in subcutaneous GBM models [308,309].

Although most recent research efforts have focused on understanding the interactions between neurons and HGGs/DMGs, it is anticipated that the relationship of gliomas with their microenvironment might be more complex. Classically, excessive neuronal activity has been thought to be controlled by specific inhibitory postsynaptic neurons (e.g., GABA-mediated); however, it was recently demonstrated that microglia play a key role in dampening the hyperexcitability of neurons [310]. It was found that microglia suppressed neuronal activity through the conversion of ATP–ADP–AMP through a cascade of purinergic receptor signalling. The subsequent conversion of AMP into adenosine by microglia (and potentially astrocytes) led to the inhibition of the A1 receptor in neurons and consequently a halt in neuronal activity [310]. Microglial dysfunction and specifically microglial activation have been observed in neurological disease settings such as neuroinflammation and epilepsy [311]. Interestingly, microglial activation has been observed in DIPG tumours; however, it is not known if they influence this novel microglial function to enhance their interaction with neurons [26]. Pharmacological targeting of A1 receptors potentially represents a new therapeutic avenue that is worth exploring further.

Apart from the local microenvironment, a rather more distant relationship has also been observed between the neuronal microenvironment and DMGs by influencing their metastasis beyond the brainstem [312]. Qin et al., through a series of biochemical and proteomic experiments, reported that neural precursor cells (NPCs) present in the lateral ventricle subventricular zone (SVZ) released a chemoattractant complex which stimulated the migration and invasion of DMG cells to the SVZ. This chemoattractant complex consisted of neurite-promoting protein pleiotrophin (PTN), heat shock protein 90B (HSP90B), secreted protein acidic and rich in cysteine (SPARC) and SPARC-like protein 1 (SPARCL1). The deletion of either factor of combination could delay the invasion of DMG cells to the SVZ of animals. Although pharmacological inhibition was not possible due to the lack of BBB-permeable HSP90 inhibitors, intracranial injection of a lentivirus expressing sh-RNA targeting the silencing of *hsp90b1* demonstrated a delay in DMG invasion in vivo. As BBB-permeable inhibitors of HSP90 have started emerging, future studies are needed to evaluate if they can inhibit DIPG invasion [313]. Pleiotrophin and midkine (MDK) belong to a family of heparin-binding cytokines influencing many physiological functions, including CNS development and immunity. Both bind to various receptors, including receptor protein tyrosine phosphatase ζ (RPTPζ), which was recently demonstrated in DMG invasion, low-density lipoprotein 1 (LRP1) and anaplastic lymphoma kinase (ALK). Currently, PTN and RPTPζ-targeted therapies are not available as anti-invasion therapeutic options. On the other hand, MDK, particularly the inhibition of its receptor, ALK, has demonstrated potential as a therapeutic option in GBMs [314]. However, it is not currently known if PTN exerts its effects through other receptors in DMGs.

## 8. Targeting of the Extracellular Matrix

Given the diffuse growth of DMGs in the brain, it is anticipated that it may influence the extracellular matrix (ECM) to promote its invasion of the brain parenchyma. The ECM is a highly organised network consisting of glycoproteins (e.g., fibronectin, laminin and tenascins), fibrous glycoproteins (collagen and laminin) and large amounts of glycosaminoglycans that interact either with proteins or hyaluronan [315]. Although the role of the ECM was mainly thought to be structural, it is now understood that it may influence cancer cell response to environmental changes, thus leading to migration and invasion. Key ECM targets such as tenascin-C (TN-C) have been recently found to be overexpressed in DMG tumours compared to normal brain tissue. In addition, knockdown experiments in primary DMG cultures have been suggestive of an essential role in DMG cell proliferation and migration [316]. Currently, TN-C has been targeted with antibodies; however, although this approach has shown some promise in subcutaneous glioma models, its efficacy has not been demonstrated in orthotopic DMG models, especially when the BBB remains intact [317]. Another potential target for DMG tumours that has recently been explored is the transmembrane receptor CD44. CD44 has been associated with the maintenance of cancer stem cell phenotype, adhesion to hyaluronan and invasion. Particularly in DMG cells, it was explicitly overexpressed in migrating and invading cells [318]. However, its effects on DMG growth and migration through therapeutic targeting has not been investigated in DMGs. Similarly to that mentioned above, TN-C blocking of CD44 with antibodies has demonstrated a reduction in subcutaneous glioma tumour growth while knocking down indicated enhanced sensitivity to cytotoxic agents [319,320]. Overall, targeting ECM components has been a relatively unexplored area of research for DMGs. In contrast, more knowledge has been accumulated for glioblastoma and recently reviewed comprehensively by Mohiuddin and Wakimoto [321]. Of particular interest would be to develop 3D bioengineered models that recapitulate the ECM to elucidate critical components involved in DIPG tumour invasion, migration and ultimately therapeutic targeting.

## 9. Conclusions

Over the past decade we have gained valuable insights into DMG biology and now recognise the role that histone and other genetic modifications play in the pathogenesis of the disease. Pre-clinically, there has been much success shown in targeting epigenetic dysregulation, cell cycle and proliferation mechanisms in DMGs. Whilst promising, these research endeavours have thus far demonstrated limited translation to effective patient treatment options. This in part may relate to the complex genomic dysregulation of DMGs and development of resistance mechanisms that will ultimately require the investigation of tailored combination therapies. Our understanding of DMG tumorigenicity has recently developed to include a role for the tumour microenvironment, highlighting further potential therapeutic targets involving regulation of the immune system and cross-talk between DMG tumour and neuronal cells. Combined, these facets underscore the potential for a more holistic approach when considering the development of innovative DMG treatment options.

## Figures and Tables

**Figure 1 cancers-13-06251-f001:**
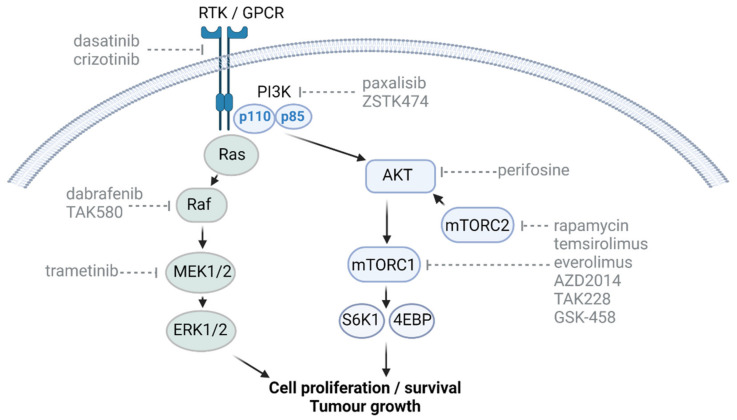
Therapeutic targets of RTK activation and PI3K/AKT/mTOR signalling pathways in DMG. Receptor tyrosine kinases (RTK) and/or G-protein-coupled receptors (GPCRs) are activated by a variety of growth factors, neurotransmitters and hormones, including insulin, brain-derived neurotropic factor, glutamate and cannabinoids (reviewed in [102]). Receptor activation recruits the intracellular association of the phosphatidylinositol 3-kinase (PI3K) regulatory (p85) and catalytic (p110) subunits. The subsequent engagement of Ras results in the activation of the Raf–mitogen-activated protein kinase kinase (MEK)–extracellular signal-regulated kinase (ERK) 1/2 pathway. Alternately, RTK-induced PI3K activation promotes AKT signalling involving the mechanistic target of the rapamycin (mTORC1/2) pathway. The triggering of these pathways initiates a variety of signals that promote DMG tumour cell proliferation, survival and tumour growth. A range of inhibitory compounds, shown in grey with dotted lines indicating specific mechanistic targets, have been tested within the setting of DMGs.

**Figure 2 cancers-13-06251-f002:**
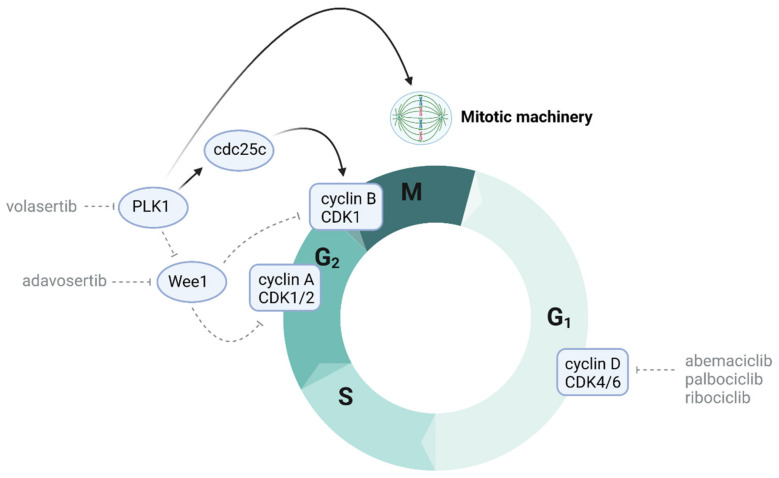
Cell-cycle-dependent therapeutic targets in DMGs. Constitutive expression and heightened activation of the cell cycle is a hallmark of DMGs. A range of inhibitory compounds, shown in grey with dotted lines indicating specific mechanistic targets, have been tested within the setting of DMGs. These include the cell cycle G1 phase, cyclin-dependant kinase 4 and 6 (CDK4/6) inhibitors abemaciclib, palbociclib and ribociclib, as well as inhibition through the targeting of molecules PLK1 and Wee1 to halt the cell cycle in the G2/M phase.

**Figure 3 cancers-13-06251-f003:**
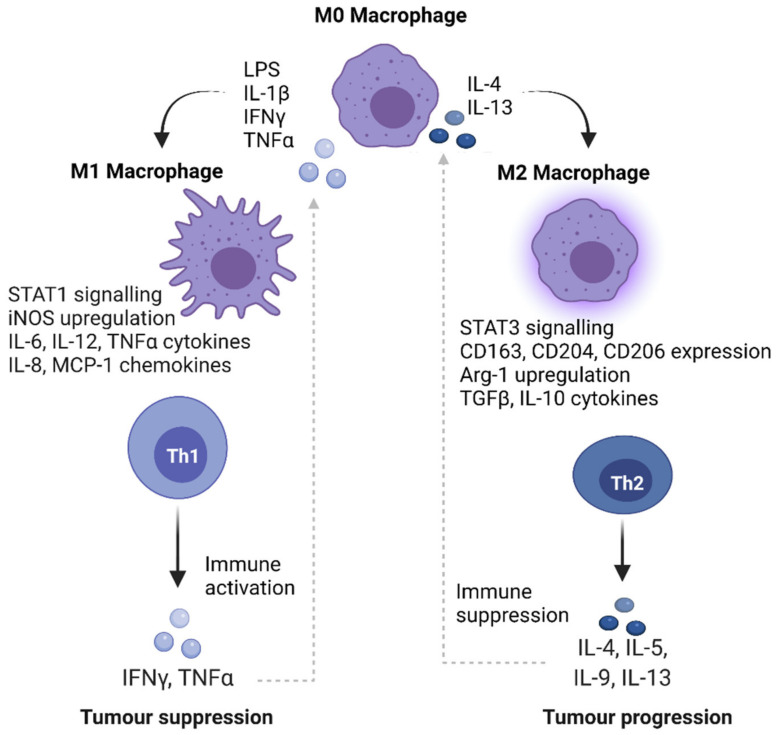
The role of macrophage phenotype and immune system signalling in DMGs. Macrophages are the predominant immune cell within the DMG microenvironment, where they exhibit a basal M0 phenotype. M0 macrophages are capable of being polarised to either a classically activated, proinflammatory M1 or an alternatively activated M2 phenotype depending entirely on the inflammatory environment encountered. Macrophages are transformed into an M1 phenotype by exposure to proinflammatory LPS, IL-1β, IFNγ or TNFα, which activates STAT1 signalling and therefore upregulates iNOS expression and the release of cytokines such as IL-6, IL-12 and TNFα along with increased production of chemokines such as IL-8 and MCP-1. Alternately, macrophages are polarised to an M2 phenotype following exposure to the cytokines IL-4 and IL-13, inducing STAT3 signalling and therefore the upregulation of cell surface markers CD 163, CD204 and CD206 as well as the release of Arg-1, TGFβ and IL-10. The activation of M1 macrophages promotes immune activation through the Th1 T cell response and subsequent release of proinflammatory IFNg and TNFa, leading to DMG tumour suppression and ongoing macrophage M1 activation. In contrast, M2 macrophages suppress the immune response through Th2 T cell activation, promoting DMG progression through establishing a tumour microenvironment which favours M2 macrophage polarisation.

**Figure 4 cancers-13-06251-f004:**
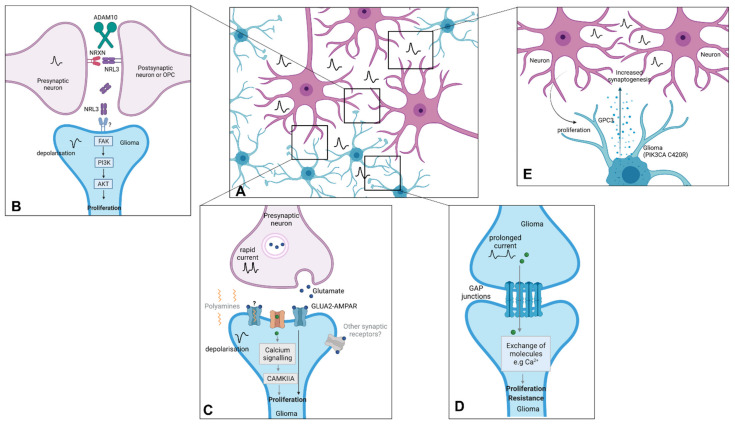
Schematic representation of established and potential interactions between the DMG and neuronal microenvironments. (**A**) DMGs form a complex network involving direct communication with neurons and glioma cells. (**B**) Typically, synaptic neuron interaction occurs through the NLG3/NRXN axis. Metalloproteinase ADAM10 cleaves NLG3 and is subsequently directed towards an unidentified receptor, leading to glioma cell proliferation through the activation of FAK and subsequently the PI3K/AKT pathway. (**C**) A rapid action potential is propagated from presynaptic neurons to DMG cells through calcium-permeable (GluA2) AMPA receptors, leading to glioma proliferation. Other synaptic receptors might be also involved in the propagation of neuron activity as well as downstream calcium signalling pathways mediated by CAMKIIA. Polyamines, short, positively charged molecules recently found to be upregulated in DMGs may also modulate the activity of GLUA2-AMPARs. (**D**) DMGs and HGGs can communicate with each other through gap junctions, promoting proliferation and resistance to radiotherapy. Long potentials were found to be mediated in DMGs through gap junctions. (**E**) Gliomas exhibiting specific PI3K mutations such as a C240R variant may increase synaptogenesis in neurons through the release of GPC3. Abbreviations: neuroligin 3 (NRLG3), neurexin (NRXN), a disintegrin and metalloproteinase domain-containing protein 10 (ADAM10), focal adhesion kinase (FAK), phosphoinositide 3-kinase (PI3K), protein kinase B (AKT), α-amino-3-hydroxy-5-methyl-4-isoxazolepropionic acid receptor (AMPAR), Ca^2+^/calmodulin-dependent protein kinase II A (CAMKIIα), phosphatidylinositol-4,5-bisphosphate 3-kinase catalytic subunit alpha (PIK3CA) and glypican 3 (GPC3).

**Table 1 cancers-13-06251-t001:** Summary of inhibition studies targeting epigenetic mechanisms in DMGs.

Epigenetic Drug Category	Drugs	Target	References
H3K27 demethylase inhibitor	GSKJ4/GSKJ1	JMJD3	[15,33,34]
EZH2 (H3K27 histone methyltransferase) inhibitor	EPZ6438 (Tazemetostat), GSK343	EZH2	[13,17,35]
BMI1 (H2AK119 ubiquitinase) inhibitor	PTC209, PTC028 and PTC596	BMI1	[36,37,38]
HDAC inhibitor	Panobinostat	HDACs	[16,39]
BET inhibitor	JQ1	BRD4	[17,19,40,41]
Curaxin	CBL0137	FACT	[21]
DNA demethylating agent	5-azacytidine	DNMTs	[42]
Epigenetic Drug Combination			
HDACi + H3K27 demethylase inhibitor	Panobinostat + GSKJ4/1	HDACs + JMJD3	[16]
HDACi + BETi	Panobinostat + JQ1	HDACs + BRD4	[40]
HDACi + CDK7i	Panobinostat + THZ1	HDACs + CDK7/9	[40]
HDACi + demethylating agent	Panobinostat + 5-azacytidine	HDACs + DNMTs	[42]
HDAC + Curaxin	Panobinostat + CBL0137	HDACs + FACT	[21]
BETi + CDK7i	JQ1 + THZ1	BRD4 + CDK7/9	[40]
BETi + EZHi	JQ1 + EPZ6468	BRD4 + EZH2	[43]
BETi + HATi	JQ1 + ICG-001	BRD4 + CBP	[44]
HDACi + H3K4me1 histone demethylase inhibitor	Corin	HDACs + LSD1	[45]

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
