# Peer review of "Therapeutic Targets in Diffuse Midline Gliomas—An Emerging Landscape"

_cancers, 2021, doi:10.3390/cancers13246251_

Round 1

Reviewer 1 Report

Authors presented comprehensive and complete review of recent updates in understsnding the molecular mechanisms of available and being under investigations  therapeutic targets in DMG. In the manuscript are many repetitions and a plethora of informations concerning other types of brain tumors, but concerning the interdisciplinary character of review and novelity of the subject, it appears to be necessary. Certain comments and suggestions are included.

  1. Targeting epigenetic mechanisms in DMG.

Excellent and complete  summary of available knowledge considering the explanation of the molecular mechanisms of action and inhibition studies, targeting of epigenetic mechanisms in DMG development and progression and  available and potential  therapies.

It should be discussed an article: Jha et al. Neuro Oncol. 2014 (12):1607-17.

  1. Targeting cell metabolism in DMG.

This part concerns mainly the dual action of metabolic enzymes as parts of metabolic pathways and as epigenetic regulators. That is a new therapeutic approach. That include the role of tumor-specific pyruvate kinase  (PKM2) and AMP-activated protein kinase (AMPK). This  enzymes in addition to their crucial role as a sensor of the  cell energy metabolism,  directly regulate gene transcription by affecting the histone phosphorylation, acetylation or methylation.

Perhaps authors will take into consideration suggested supplementations:

The PKM2 regulates the rate-limiting step of glycolysis that shifts the glucose metabolism from the respiratory chain to lactate production in tumor cells and is crucial for cancer development and progression. It should be mentioned that recently, targeting PKM2 through its regulatory microRNAs, long non-coding RNAs (lncRNAs), and circular RNAs (circRNAs) has gathered increasing interest.( Puckett et al.. Int J Mol Sci. 2021;22(3):1171) . It should be interesting to add that  PKM expression, but not activity, is regulated in a grade-specific manner in glioma, but changes in both PKM activity and PKM2 expression contribute to growth of GBM. (Mukherjee et al. PLoS One. 2013;8(2):e57610).

It is established that PKM1/2 knockdown activated AMPK signaling (Prakasam et al. J Biol Chem. 2017 ;292(37):15561-15576.). AMPK activates energy-producing pathways and inhibits energy-consuming processes as cell growth and proliferation. AMPK activity opposes tumor development and progression also in part by regulating inflammation and metabolism  (Li et al. Oncotarget. 2015;6(10):7365-7378).

Manipulation of the  switch  between the glycolysis and oxidative phosphorylation pathways is a target to anticancer therapy. Pyruvate dehydrogenase kinase (PDK1) activates the entry of acetylo-CoA to citric acid cycle and oxidative phosphorylation. Simultaneous AMPK activation and PDK1 suppression is able to repress the glycolytic phenotype  and suppress the DIPG in vivo and in vitro. Aso AMPK induction to suppress glycolysis through mTOR signalling  was able to decrease  DMG progression.

Combination therapy was found to act through inhibition of PI3K/AKT/mTOR pathway, HSP90 and activation of AMPK by inhibiting mitochondria adenine nucleotide translocase (ANT) with a novel anti-cancer compound  PENAO upon co-treatment with mTOR inhibitor temsirolimus in DIPG neurosphere cultures (Tsoli et al. Oncotarget. 2018 Jan 8;9(7):7541-7556).   

Again # 3. Targeting cellular signalling pathways in DMG

Autors  presented an extensive description of the  cancer signalling pathways involved in GBM and DBM development and progression, including  PI3K/AKT/mTORC1 and mTORC2 pathway, TGFβ pathway involving the TGFβ and BMP -mediated signalling with recognition of the the ubiquitin-proteasome system (UPS), components , including SMAD and their interacted proteins in the regulation of the TGFβ family signalling. The specific inhibitors and their anticancer activity as well as clinical trials were also discussed.

Certain comments are proposed:

It has to be pointed out that the cancer signalling pathways activated by the various growth factor receptors acts through the specific receptor tyrosine kinases.  mTOR is activated by many receptors including growth factors, hormones (insuline) and neurotransmitters through  the specific receptor tyrosine kinases and  G-proteins coupled receptors.  This should be included in Fig.1.

In addition to presented description is  also worth to explain in simple way that mTOR is serine/threonine kinase that form two distinct complexes, mTORC1 and mTORC2 that  differ in the protein components, substrate specificity and regulation. mTORC1 contains a protein termed Raptor which allows it to phosphorylate specific substrates. mTORC2 contains Rictor in place of Raptor and as a 'dual-specificity' protein kinase phosphorylating  tyrosine as well as serine/threonine sites  of a distinct set of substrates (Wang, and  Proud, Cell Res 26, 1–2 (2016).

To make the article more friendly for readers It would be beneficial to include list of used abbrevations e.g. grow factor receptors and corresponding  mediators of signaling pathways.

Also a part of the  investigations concerning use of various inhibitors in  treatment and references could be presented as Tables , similarly to Table 1.

  1. Targeting the cell cycle and DNA repair mechanisms in DMG.

Autors showed that targeting DNA repair using PARP inhibitors may be a beneficial strategy for a homologous recombination -deficient DMG . Pointed out is beneficial application of combination  therapy comprising the  inhibitors of CDK4/6 and PI3K/AKT/mTOR pathway in DMGs therapy.The cell cycle progression, 1-S and G2-M checkpoint activation and arrest with particular role of a  checkpoint kinases  ATM and CHK1/2 and PLK1 that activates CDK1/cyclin B complex and causes Wee1 degradation,  were also discussed as the target for  DMG therapy.

  1. Activating the immune response as a potential therapeutic option in DMG

The origin of TAM in DMG was described as well as the macrophage  (M0) plasticity for polarization into proinflammatory, with anticancer activity M1 (STAT 1 pathway)and immunosuppresive M2  (STAT 3) phenotypes . It has been shown that macrophages  and microglia from normal brain regions and TAMs  from GMb and DMG , could not be categorised into individual polarisation phenotypes and resamble M0 type that corresponds to low inflammatory state. The transition of  that TAMs  towards an M1 inflammatory phenotype may  promote  the lymphocyte Th1  cytokine response, resulting in tumor suppression  Inhibition of STAT 3 , however abrogate M2 polarization,  creates ambigous results.   Similarly the effecting of PLK1 and TAM TLR-4 signalling was described for other cancer types without results for DMG. As well  as the other approach concerning the use of combined the polyamine synthesis and transport inhibitors .

Pointed out is the high sensitivity of DMG to HDAC inhibition potentiating the activation of the M1macrophages response . Also data are presented that EZH2 inhibition may exert its clinical benefit within the setting of DIPG.

The inhibitory effect of M2-like phenotype macrophages  on cytotoxic CD8+ T  cells subtype was  presented . GBM tumor cell kynurenine production activates the AHR on TAMs, decreasing NFKβ signalling and inducing the M2 phenotype.  The increased expression of Arg-1 by M2 macrophages directly suppress T-cell function within the microenvironment through the arginase-mediated depletion of arginine.

T cell transfer therapy was presented as alternative.  The methods and treatments optymalizing that kind of DMG therapy were described. Particularly interesting is of the checkpoint molecule B7-H3 that is high on DMG and  CAR T cells directed at B7-H3 are capable of crossing the BBB and producing sufficient tumor killing amounts of cytotoxic cytokines and in vitro in coculture with DMG cells.

It would be interesting to include the short characteristic of macrophage partners in cancer micro environment, namely natural killers (NK) lymphocytes.  (Price et al. The Lancet, 2021.69; 103453).

  1. Targeting of neuronal cell-DMG interactions

The presence of functional synapses between DMG cells and neurons is considered. The   neuronal activity could stimulate the growth of HGG and DMG cells through the secretion of synaptic factors such as neuroligin-3 (NLG3) and the  Focal Adhesion Kinase (FAK) and downstream signalling through the PI3K/AKT pathway. The role of metalloproteinase ADAM10 in that pathway is shown.  The possibile role of the polyamines may play a protective role in DMG at the local level where calcium-permeable AMPARs mediate the propagation of action potentials .

Considering the migration and invasion of the DMG cells  it was described the new chemoattractant complex that is released by the neural precursor cells (NPCs) and consists of fiew components necessary for its activity ,that is a new target for DGM therapy.

Perhaps should be mentioned the emerging therapeutic approaches for GBM that target or utilize its unique extracellular matrix  (Mohiuddin  and Wakimoto . Am J Cancer Res. 2021;11(8):3742-3754).

Author Response

Authors presented comprehensive and complete review of recent updates in understsnding the molecular mechanisms of available and being under investigations  therapeutic targets in DMG. In the manuscript are many repetitions and a plethora of informations concerning other types of brain tumors, but concerning the interdisciplinary character of review and novelity of the subject, it appears to be necessary. Certain comments and suggestions are included.

Targeting epigenetic mechanisms in DMG.

Excellent and complete  summary of available knowledge considering the explanation of the molecular mechanisms of action and inhibition studies, targeting of epigenetic mechanisms in DMG development and progression and  available and potential  therapies.

It should be discussed an article: Jha et al. Neuro Oncol. 2014 (12):1607-17.

This reference, which validates the findings of Sturm et al., has now been incorporated into the paragraph on DNA methylation on Page 10 of the manuscript.

Targeting cell metabolism in DMG.

This part concerns mainly the dual action of metabolic enzymes as parts of metabolic pathways and as epigenetic regulators. That is a new therapeutic approach. That include the role of tumor-specific pyruvate kinase  (PKM2) and AMP-activated protein kinase (AMPK). This  enzymes in addition to their crucial role as a sensor of the  cell energy metabolism,  directly regulate gene transcription by affecting the histone phosphorylation, acetylation or methylation.

Perhaps authors will take into consideration suggested supplementations:

The PKM2 regulates the rate-limiting step of glycolysis that shifts the glucose metabolism from the respiratory chain to lactate production in tumor cells and is crucial for cancer development and progression. It should be mentioned that recently, targeting PKM2 through its regulatory microRNAs, long non-coding RNAs (lncRNAs), and circular RNAs (circRNAs) has gathered increasing interest.( Puckett et al.. Int J Mol Sci. 2021;22(3):1171) . It should be interesting to add that  PKM expression, but not activity, is regulated in a grade-specific manner in glioma, but changes in both PKM activity and PKM2 expression contribute to growth of GBM. (Mukherjee et al. PLoS One. 2013;8(2):e57610).

It is established that PKM1/2 knockdown activated AMPK signaling (Prakasam et al. J Biol Chem. 2017 ;292(37):15561-15576.). AMPK activates energy-producing pathways and inhibits energy-consuming processes as cell growth and proliferation. AMPK activity opposes tumor development and progression also in part by regulating inflammation and metabolism  (Li et al. Oncotarget. 2015;6(10):7365-7378).

Manipulation of the  switch  between the glycolysis and oxidative phosphorylation pathways is a target to anticancer therapy. Pyruvate dehydrogenase kinase (PDK1) activates the entry of acetylo-CoA to citric acid cycle and oxidative phosphorylation. Simultaneous AMPK activation and PDK1 suppression is able to repress the glycolytic phenotype  and suppress the DIPG in vivo and in vitro. Aso AMPK induction to suppress glycolysis through mTOR signalling  was able to decrease  DMG progression.

Combination therapy was found to act through inhibition of PI3K/AKT/mTOR pathway, HSP90 and activation of AMPK by inhibiting mitochondria adenine nucleotide translocase (ANT) with a novel anti-cancer compound  PENAO upon co-treatment with mTOR inhibitor temsirolimus in DIPG neurosphere cultures (Tsoli et al. Oncotarget. 2018 Jan 8;9(7):7541-7556).  

We thank the reviewer for the comprehensive suggestion. Some of the suggested detail is already covered in other sections of the manuscript, but we have amended the text of the manuscript at page 12 with the following:

“PKM2 catalyzes the rate-limiting step of glycolysis shunting glucose metabolism away from oxidative phosphorylation  towards anaerobic glycolysis and lactate production in tumor cells, a key feature during cancer development and progression. Depletion of PKM2 with regulatory microRNAs (miRNA), long non-coding RNAs (lncRNAs), and circular RNAs (circRNAs) have yielded promising results in in vitro models of GBM [83]. PKM2 expression, but not activity, is regulated in a grade-specific manner in gliomas, but changes in both PKM activity and PKM2 expression contribute to growth of GBM. Knockdown of PKM1/2 activated AMP-activated protein kinase (AMPK1) and suppressed viability in lung carcinoma cell lines [84]. AMPK1 plays an important role in maintaining H3K27 methylation deficiency. AMPK1 has been shown to directly target EZH2, disrupting EZH2-dependent methylation of H3K27 and consequently PRC2 activity [13, 85].”

Again # 3. Targeting cellular signalling pathways in DMG

Autors  presented an extensive description of the  cancer signalling pathways involved in GBM and DBM development and progression, including  PI3K/AKT/mTORC1 and mTORC2 pathway, TGFβ pathway involving the TGFβ and BMP -mediated signalling with recognition of the the ubiquitin-proteasome system (UPS), components , including SMAD and their interacted proteins in the regulation of the TGFβ family signalling. The specific inhibitors and their anticancer activity as well as clinical trials were also discussed.

Certain comments are proposed:

It has to be pointed out that the cancer signalling pathways activated by the various growth factor receptors acts through the specific receptor tyrosine kinases. 

We thank the reviewer for this suggestion, and have now added to the RTK section on page 14 of the manuscript, as follows:

“Growth factor receptors are receptor tyrosine kinases (RTKs) and facilitate their downstream activity through activation of the specific RTK.”

mTOR is activated by many receptors including growth factors, hormones (insuline) and neurotransmitters through  the specific receptor tyrosine kinases and  G-proteins coupled receptors. 

A sentence and reference has been added in the PI3K/mTOR section in the text on page 14 of the manuscript, as follows:

“The PI3K/AKT/mTOR intracellular signalling pathway is important in regulating the cell cycle, with mTOR activity a common target for cancer therapeutics, being frequently upregulated across multiple cancer types. Growth factors, neurotransmitters and hormones are all able to activate the mTOR pathway through their specific RTKs and G-protein coupled receptors (Takei & Nawa, 2014).”

This should be included in Fig.1.

Figure 1 has been amended to include GPCR and the text in the legend amended, as follows:

“Receptor tyrosine kinases (RTK) and/or G-protein coupled receptors (GPCR) are activated by a variety of growth factors, neurotransmitters and hormones, including insulin, brain-derived neurotropic factor, glutamate, and cannabiniods (reviewed in Takai & Nawa, 2014). Receptor activation recruits the intracelluar association of the Phosphatidylinositol 3-kinase (PI3K) regulatory (p85) and catalytic (p110) subunits…”

In addition to presented description is  also worth to explain in simple way that mTOR is serine/threonine kinase that form two distinct complexes, mTORC1 and mTORC2 that  differ in the protein components, substrate specificity and regulation. mTORC1 contains a protein termed Raptor which allows it to phosphorylate specific substrates. mTORC2 contains Rictor in place of Raptor and as a 'dual-specificity' protein kinase phosphorylating  tyrosine as well as serine/threonine sites  of a distinct set of substrates (Wang, and  Proud, Cell Res 26, 1–2 (2016).

We thank the reviewer for the suggestion, and have amended the text and added the requested reference on page 15 of the manuscript, as follows:

“mTOR is a serine/threonine protein kinase  composed of two distinct complexes which differ in components, substrate specificity and downstream regulation; mTORC1, composed of mTOR, raptor, GβL and deptor, and mTORC2, composed of mTOR, rictor, GβL, PRR5, deptor and SIN1 [105] (Wang and Proud, 2016 (PMID: 26846308).”

To make the article more friendly for readers It would be beneficial to include list of used abbrevations e.g. grow factor receptors and corresponding  mediators of signaling pathways.

The abbreviation list is extensive, and would cover 5 pages of text. We have therefore, at this stage decided not to include this in the manuscript.

Also a part of the  investigations concerning use of various inhibitors in  treatment and references could be presented as Tables , similarly to Table 1.

We thank the reviewer for the suggestion, but feel that tabulating the inhibitor findings would detract from, rather than add to, the text and data presented in the manuscript.

Targeting the cell cycle and DNA repair mechanisms in DMG.

Autors showed that targeting DNA repair using PARP inhibitors may be a beneficial strategy for a homologous recombination -deficient DMG . Pointed out is beneficial application of combination  therapy comprising the  inhibitors of CDK4/6 and PI3K/AKT/mTOR pathway in DMGs therapy.The cell cycle progression, 1-S and G2-M checkpoint activation and arrest with particular role of a  checkpoint kinases  ATM and CHK1/2 and PLK1 that activates CDK1/cyclin B complex and causes Wee1 degradation,  were also discussed as the target for  DMG therapy.

Activating the immune response as a potential therapeutic option in DMG

The origin of TAM in DMG was described as well as the macrophage  (M0) plasticity for polarization into proinflammatory, with anticancer activity M1 (STAT 1 pathway)and immunosuppresive M2  (STAT 3) phenotypes . It has been shown that macrophages  and microglia from normal brain regions and TAMs  from GMb and DMG , could not be categorised into individual polarisation phenotypes and resamble M0 type that corresponds to low inflammatory state. The transition of  that TAMs  towards an M1 inflammatory phenotype may  promote  the lymphocyte Th1  cytokine response, resulting in tumor suppression  Inhibition of STAT 3 , however abrogate M2 polarization,  creates ambigous results.   Similarly the effecting of PLK1 and TAM TLR-4 signalling was described for other cancer types without results for DMG. As well  as the other approach concerning the use of combined the polyamine synthesis and transport inhibitors .

Pointed out is the high sensitivity of DMG to HDAC inhibition potentiating the activation of the M1macrophages response . Also data are presented that EZH2 inhibition may exert its clinical benefit within the setting of DIPG.

The inhibitory effect of M2-like phenotype macrophages  on cytotoxic CD8+ T  cells subtype was  presented . GBM tumor cell kynurenine production activates the AHR on TAMs, decreasing NFKβ signalling and inducing the M2 phenotype.  The increased expression of Arg-1 by M2 macrophages directly suppress T-cell function within the microenvironment through the arginase-mediated depletion of arginine.

T cell transfer therapy was presented as alternative.  The methods and treatments optymalizing that kind of DMG therapy were described. Particularly interesting is of the checkpoint molecule B7-H3 that is high on DMG and  CAR T cells directed at B7-H3 are capable of crossing the BBB and producing sufficient tumor killing amounts of cytotoxic cytokines and in vitro in coculture with DMG cells.

It would be interesting to include the short characteristic of macrophage partners in cancer micro environment, namely natural killers (NK) lymphocytes.  (Price et al. The Lancet, 2021.69; 103453).

We thank the reviewer for the suggestion and have amended page 26 of the manuscript to include the suggested comment and reference, as per the following:

“However, the immune environment in DMG is largely non-inflammatory, with a low adaptive immune response and decreased infiltration of Natural Killer (NK) cells (reviewed in [224]). Of the immune cells present, bone marrow-derived macrophages (BMDM), over microglia, are the predominant tumor-associated macrophages/microglia (TAM) subpopulation in DMG [25] and it has been established that human DMG samples express high levels of the pan-macrophage markers CD11b, CD45 and CD68 concurrent with increased expression of the M2 marker CD163 [225], confirming that the high expression of chemo-attractants results in the enhanced infiltration of, in particular, alternatively-activated macrophages within this disease setting. Despite this knowledge, therapeutic targeting of non-tumor infiltrating cell types, including TAMs, has only recently began to be investigated.”

Targeting of neuronal cell-DMG interactions

The presence of functional synapses between DMG cells and neurons is considered. The   neuronal activity could stimulate the growth of HGG and DMG cells through the secretion of synaptic factors such as neuroligin-3 (NLG3) and the  Focal Adhesion Kinase (FAK) and downstream signalling through the PI3K/AKT pathway. The role of metalloproteinase ADAM10 in that pathway is shown.  The possibile role of the polyamines may play a protective role in DMG at the local level where calcium-permeable AMPARs mediate the propagation of action potentials .

Considering the migration and invasion of the DMG cells  it was described the new chemoattractant complex that is released by the neural precursor cells (NPCs) and consists of fiew components necessary for its activity ,that is a new target for DGM therapy.

Perhaps should be mentioned the emerging therapeutic approaches for GBM that target or utilize its unique extracellular matrix  (Mohiuddin  and Wakimoto . Am J Cancer Res. 2021;11(8):3742-3754).

We thank the reviewer for this fantastic suggestion and have now included an additional section to address the therapeutic targeting of the extracellular matrix (ECM).  Unfortunately, only a couple of published studies have attempted to cover some aspects of ECM in DMG biology, such as Tenascin and CD44. We have provided information on these two targets for both DMGs and glioblastoma tumours. Given this is a more DMG related review article, we have included the suggested reference by Mohiuddin and Wakimoto and prompted the readers to look within for more glioblastoma related research. We do understand that this is an unexplored area of research that requires more attention.

The amendments are on page 39 of the manuscript, as follows:

“Targeting of the extracellular matrix

Given the diffuse growth of DMGs in the brain, it is anticipated that it may influence the extracellular matrix (ECM) to promote its invasion in the brain parenchyma. The ECM is a highly organised network consisting of glycoproteins (e.g. fibronectin, laminin and tenascins) and fibrous glycoproteins (collagen and laminin), as well as large amounts of glycosaminoglycans interacting either with proteins or hyaluronan[300]. Although the role of the ECM was mainly thought to be structural, it is now understood that it may influence cancer cell response to environmental changes leading thus to migration and invasion. Key ECM targets such as tenascin-C (TN-C) have been recently found to be overexpressed in DMG tumours compared to normal brain tissue. In addition, knockdown experiments in primary DMG cultures have been suggestive of an essential role in DMG cell proliferation and migration[301]. Currently, TN-C has been targeted with antibodies; however, although this approach has shown some promise in subcutaneous glioma models, its efficacy has not been demonstrated in DMG orthotopic models, especially when the BBB remains intact[302]. Another potential target for DMG tumours recently explored is the transmembrane receptor CD44. CD44 has been associated with the maintenance of cancer stem cell phenotype, adhesion to hyaluronan and invasion. Particularly in DMG cells, it was explicitly overexpressed in migrating and invading cells[303]. However, its effects on DMG growth and migration through therapeutic targeting has not been investigated in DMG. Similarly to above mentioned TN-C blocking of CD44 with antibodies has demonstrated a reduction in subcutaneous glioma tumour growth while knocking down indicated enhanced sensitivity to cytotoxic agents[304, 305]. Overall, targeting ECM components has been a relatively unexplored area of research for DMG. In contrast, more knowledge has been accumulated for glioblastoma and recently reviewed comprehensively by Mohiuddin and Wakimoto[306]. Of particular interest would be to develop 3D bioengineered models that recapitulate the ECM to elucidate critical components involved in DIPG tumour invasion, migration and ultimately therapeutic targeting.”

Reviewer 2 Report

The review is very exhaustive and well written on an emering topic especially in light of new targeted therapies.

I have no specific concerns or changes to suggest.

Author Response

Reviewer 2.

The review is very exhaustive and well written on an emering topic especially in light of new targeted therapies. I have no specific concerns or changes to suggest.

We thank the reviewer.

Reviewer 3 Report

The authors have done a commendable work by organizing a comprehensive review which summarizes the understanding of Diffuse Midline Gliomas.

The epigenetics of DMGs have been discussed in detail and provides very pertinent information.Besides,topics like targeting cellular signaling pathways,targetings the cell cycle and DNA repair mechanisms are also covered in detail.

The most interesting and informative topics deals with activating the immune response as a potential therapeutic modality for DMGs.

The review article merits to be published in your esteemed journal.

Author Response

Reviewer 3.

The authors have done a commendable work by organizing a comprehensive review which summarizes the understanding of Diffuse Midline Gliomas.

The epigenetics of DMGs have been discussed in detail and provides very pertinent information.Besides,topics like targeting cellular signaling pathways,targetings the cell cycle and DNA repair mechanisms are also covered in detail. The most interesting and informative topics deals with activating the immune response as a potential therapeutic modality for DMGs. The review article merits to be published in your esteemed journal.

We thank the reviewer.